# Transcription factor Znf2 coordinates with the chromatin remodeling SWI/SNF complex to regulate cryptococcal cellular differentiation

Jianfeng Lin[1], Youbao Zhao [1], Aileen R. Ferraro [1], Ence Yang[2], Zachary A. Lewis [1,3,4] & Xiaorong Lin [1,3,5]*

Cellular differentiation is instructed by developmental regulators in coordination with chromatin remodeling complexes. Much information about their coordination comes from studies in the model ascomycetous yeasts. It is not clear, however, what kind of information that can be extrapolated to species of other phyla in Kingdom Fungi. In the basidiomycete *Cryptococcus neoformans*, the transcription factor Znf2 controls yeast-to-hypha differentiation. Through a forward genetic screen, we identified the basidiomycete-specific factor Brf1. We discovered Brf1 works together with Snf5 in the SWI/SNF chromatin remodeling complex in concert with existent Znf2 to execute cellular differentiation. We demonstrated that SWI/SNF assists Znf2 in opening the promoter regions of hyphal specific genes, including the *ZNF2* gene itself. This complex also supports Znf2 to fully associate with its target regions. Importantly, our findings revealed key differences in composition and biological function of the SWI/SNF complex in the two major phyla of Kingdom Fungi.

[1] Department of Microbiology, University of Georgia, Athens, GA 30602, USA. [2] Department of Microbiology & Infectious Disease Center, School of Basic Medical Sciences, Peking University Health Science Center, 100191 Beijing, China. [3] Department of Plant Biology, University of Georgia, Athens, GA 30602, USA. [4] Department of Genetics, University of Georgia, Athens, GA 30602, USA. [5] Department of Infectious Diseases, University of Georgia, Athens, GA 30602, USA. *email: Xiaorong.Lin@uga.edu

Cellular differentiation allows genetically identical cells to exhibit distinct phenotypes. Although functional and morphological differences are most pronounced in higher eukaryotes, cellular differentiation is a universal phenomenon, often directed by differential gene expression. In eukaryotes, nucleosomes serve as a general barrier preventing transcription and specific regulatory mechanisms make certain genomic DNA regions accessible to transcription factors and RNA polymerase II. The ATP-dependent SWItch/Sucrose Non-Fermentable (SWI/SNF) chromatin remodeling family mainly facilitates *in cis* sliding and/or *in trans* ejecting of nucleosomes on DNA[1], allowing transcription activation or repression. Through modulating gene expression, the SWI/SNF family of complexes are critical to a variety of cellular processes including stemness and differentiation.

The SWI/SNF complex is composed of 12 subunits in *Saccharomyces cerevisiae* and 11–15 subunits in humans[2,3]. However, most human SWI/SNF subunits have several isoforms, permitting dozens of combinatorial assemblies and a spectrum of related complexes[4]. It is, therefore, challenging to attribute observed phenotypes based on a mutation of a particular subunit to the function of a specific complex. Here, we use the term family when we discuss the SWI/SNF family of complexes. In *S. cerevisiae* and *Schizosaccharomyces pombe*, there are only two complexes in the SWI/SNF family: the SWI/SNF complex and the RSC (remodeling structure of chromatin) complex. These two complexes have their distinct catalytic and accessory subunits while sharing some common subunits[5,6]. The relative simplicity in these model yeasts has facilitated investigation of the molecular functions of SWI/SNF complexes and individual subunits. It is unclear, however, what functions and features of the SWI/SNF complexes established in these two model ascomycete yeasts can be extrapolated to other eukaryotic species, including other distantly related fungal species.

In contrast to these model yeasts, *Cryptococcus neoformans* belongs to a different phylum in Kingdom Fungi: Basidiomycota. Basidiomycetes diverged from ascomycetes about one billion years ago. They share key features with higher eukaryotes that are absent from the model yeasts. For instance, >90% of cryptococcal protein-coding genes contain multiple introns. Epigenetic regulation, such as RNAi and DNA methylation, plays important roles in cryptococcal biology[7–10]. *C. neoformans* can exist in multiple morphotypes and morphogenesis is associated with its pathogenicity[11]. For instance, yeasts and spores are infectious and virulent[12,13]; titan cells are proposed to be dormant and stress-resistant in hosts[14,15]; pseudohyphae and hyphae are attenuated in virulence in mammalian hosts[16]. In the environment, however, hyphae are an integral part of its life cycle and confer cryptococcal resistance to its natural predators like soil amoeba[17].

The yeast-to-hypha transition is the best-understood cellular differentiation process in *C. neoformans*. During bisexual reproduction, **a** and α yeast partners conjugate after activating the pheromone-sensing cascade, and the resulting zygote differentiates to hyphae. In the absence of a compatible mating partner, the fungus can undergo similar cellular differentiation events[18], with the exception that the non-self-recognition system is dispensable[19,20]. The zinc-finger transcription factor, Znf2, is an essential regulator for hyphal growth[16,19–21]. Deletion of *ZNF2* confines cryptococcal cells to the yeast form and overexpression of *ZNF2* drives filamentation regardless of growth conditions[16,21]. It is unknown whether chromatin remodeling factors coordinate with Znf2 to control the yeast-hypha differentiation in this basidiomycete.

The ATP-dependent chromatin remodeling SWI/SNF family complexes were initially discovered in *Saccharomyces* through genetic screens for mating-type switching or sucrose metabolism factors[22,23]. Here, through a forward genetic screen in a *ZNF2* overexpression strain in *C. neoformans*, we identified two genes

*SNF5* and *BRF1* that are essential for hyphal differentiation even when Znf2 protein is produced. Snf5 is a conserved core subunit in the SWI/SNF complex, while Brf5 is a novel basidiomycete-specific protein. We discovered that Brf1 works together with Snf5 in the SWI/SNF complex. We further demonstrated that Brf1 is essential for transcriptional induction of *ZNF2* and is required for Znf2's full association to the promoter regions of its downstream target genes, including the *ZNF2* gene itself. Furthermore, the promoter region of *ZNF2* and its downstream targets important for filamentation become transcriptionally inaccessible in the absence of *BRF1* or *SNF5*. To our knowledge, this is the first identification and functional characterization of a phylum-specific subunit in the SWI/SNF complex in basidiomycetes. Our findings also revealed major differences in composition and biological functions of the SWI/SNF and the RSC complexes between the two major phyla in the Kingdom Fungi.

## Results

**Nuclear Znf2 fails to induce hyphae in insertional mutants**. We previously demonstrated the key function of *ZNF2* in regulating yeast-to-hypha transition. Here, we employed a reporter strain to identify Znf2's partners through a forward genetic screen. In this reporter strain, the native *ZNF2* gene is deleted and an ectopic copy of mCherry-fused *ZNF2* is expressed under the control of an inducible promotor of a copper transporter *CTR4*. This reporter strain switched from yeasts to hyphae when *ZNF2* was induced in the presence of the copper chelator bathocuproinedisulfonic acid (BCS) (Fig. 1a), as expected based on our previous studies[21,24]. The production of Znf2 can be monitored through the nuclear-localized mCherry signal (Fig. 1a). Here we used insertional mutagenesis through *Agrobacterium*-mediated transformation (AMT) and visually screened 88,000 T-DNA insertional mutants made in this reporter strain for smooth yeast colonies on filamentation-inducing V8 agar medium containing BCS (Fig. 1b). Eighty-four mutants defective in filamentation were isolated (strains AMT001-AMT084, Supplementary Data 1).

Fifty-eight mutants showed nuclear mCherry-Znf2 while 26 had altered subcellular distribution of the fluorescence signal. The mutants with mis-localized Znf2 signal might be defective in Znf2 trafficking, post-translational modification, and/or stability. Here, we set to identify factors that are required for nuclear-localized Znf2 protein to exert its function. Among these 58 mutants, 25 candidates showed only yeast cellular morphology and nuclear mCherry-Znf2 signal when cultured in YPD + BCS-inducing condition (Supplementary Data 1). These 25 mutants were chosen for further analyses.

Multiple T-DNA insertions and cryptic mutations could occur during AMT[25,26]. To determine if the T-DNA insertion is genetically linked to the non-filamentous phenotype, we back-crossed each of these 25 mutants (α) to the wild-type congenic mating partner XL280**a**, micro-dissected basidiospores, and analyzed the phenotypes of the meiotic progeny. The genetic linkage analyses indicate that 8 out of the 25 mutants likely harbor a sole T-DNA insertion that is genetically linked to their non-filamentous phenotype on BCS media (Fig. 1c).

**Two candidates were identified as filamentation activators**. To identify the genes affected by the T-DNA insertions in the selected eight mutants, we used the genome sequencing approach developed by Dr. Alspaugh's group[26]. Eight insertion sites were identified from the genome sequences of the progeny of the eight insertional mutants where only the HYG[R] (T-DNA) is present (Supplementary Table 1). Two singleton insertions were not pursued further as they could be the result of big chromosomal fragment deletion or chromosomal rearrangement[26]. The remaining six paired insertions

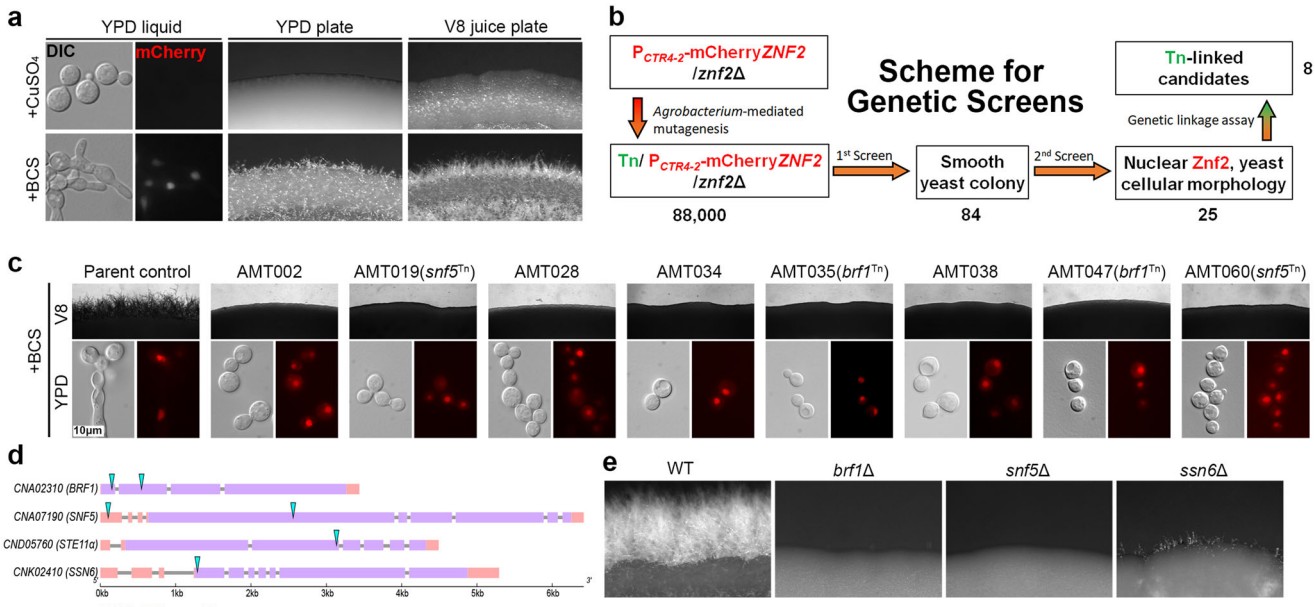

**Fig. 1** *BRF1* and *SNF5* are essential factors for filamentation. **a** Phenotypes of the parental reporter strain P$_{CTR4-2}$-mCherry-*ZNF2/znf2*Δ used in the forward genetic screen. The reporter strain was cultured under Znf2-inducing conditions (YPD + BCS liquid overnight, YPD + BCS, and V8 + BCS agar medium for 2 days) or Znf2-suppressing conditions (YPD + CuSO$_4$, V8 + CuSO$_4$). The fluorescence signal of mCherry-Znf2 was detected in the nucleus under inducing conditions. A fluffy colony edge reflects filamentous growth while a smooth edge reflects yeast growth. **b** The scheme for the genetic screens to identify factors important for filamentation when Znf2 is produced and localized in the nucleus. **c** The eight selected insertional mutants were cultured on V8 + BCS medium for 2 days or in YPD + BCS medium overnight. **d** The genetic loci disrupted by the T-DNA insertions with paired sequences. The cyan triangles indicate the T-DNA insertion sites. **e** WT XL280 and the independent targeted gene deletion mutants *brf1*Δ, *snf5*Δ, and *ssn6*Δ were cultured on V8 medium for 7 days

revealed four possible genetic loci (Fig. 1d). *CND05760* encodes the Ste11α MAP kinase in the pheromone-sensing cascade, but it is not essential for filamentation[20]. We knocked out the other three identified genes *CNK02410*, *CNA02310*, and *CNA07190* in wild-type XL280. Deletion of *CNK02410* (*SSN6*), which encodes a general transcriptional co-repressor, reduced filamentation in the wild-type (WT) background (Fig. 1e). *CNA02310* and *CNA07190* were recovered twice from independent insertions (Fig. 1d), and independent targeted deletion of these two genes in the WT background nearly abolished filamentation, similar to the insertional mutants (Fig. 1c, e). Thus, *CNA02310* and *CNA07190* are essential for yeast-hypha transition.

*CNA07190* encodes Snf5 (1784 aa), a core subunit of the conserved SWI/SNF complex. Snf5 is critical for cellular differentiation in all organisms tested, including ascomycetous and basidiomycetous fungi[27–32]. In *C. neoformans*, a bilateral *snf5*Δα × *snf5*Δ**a** cross yielded no mating hyphae or fruiting body in the serotype A H99 strain background[25]. Consistently, we found that a bilateral *snf5*Δα × *snf5*Δ**a** cross in the serotype D XL280 background yielded no mating hyphae (Fig. 2a). Thus, the function of Snf5 is conserved in both serotypes. Besides Snf5, no other subunits in the SWI/SNF complex have been identified or characterized in *C. neoformans*.

*CNA02310* encodes an uncharacterized novel protein (1033 aa). The predicted protein has an AT-rich interacting domain (ARID) but no other recognizable domains. Because it is basidiomycete specific (Supplementary Fig. 1), we named it *BRF1* (Basidiomycete-specific Regulator of Filamentation 1).

**Brf1 functions in the same biological process as Snf5.** As mentioned earlier, deletion or T-DNA disruption of *SNF5* or *BRF1* abolished or nearly abolished self-filamentation (Fig. 1e). Unilateral crosses between the *snf5*Δ or the *brf1*Δ mutant with a

non-self-filamentous reference strain on V8 medium produced fewer mating hyphae (Supplementary Fig. 2a), likely due to reduced cell fusion efficiency (Supplementary Fig. 2b). Bilateral crosses between the mutant partners (*snf5*Δα × *snf5*Δ**a** or *brf1*Δα × *brf1*Δ**a**) did not produce any aerial mating hyphae or sexual spores (Fig. 2a). As bisexual mating or self-filamentation on V8 medium is mostly driven by the pheromone signaling cascade, we decided to examine the impact of these mutations on filamentation on V8 medium containing 500 μM of copper where self-filamentation is independent of the non-self-recognition system[19]. Neither *brf1*Δ nor *snf5*Δ filamented under this condition (Fig. 2a), confirming the similarly critical role of Brf1 and Snf5 in filamentation.

Brf1 and Snf5 also share similar functions in other assays. Both *brf1*Δ and *snf5*Δ showed slightly increased sensitivity to osmotic stress (Fig. 2b). Both mutants grew similarly to WT in other conditions tested. The increased sensitivity to osmotic stress in *brf1*Δ might be due to the decreased transcript level of genes involved in anion and ion transportation (Supplementary Data 2), which was not observed in the *znf2*Δ mutant. *SNF5* is known to be required for normal growth on media with sucrose or raffinose as the sole carbon source[25]. Here we found that the *brf1*Δ mutant showed a similar defect in growth on the raffinose medium as the *snf5*Δ mutant (Fig. 2c, d). Neither the *brf1*Δ mutant nor the *snf5*Δ mutant showed any obvious defect in growth when glucose was used as the carbon source (Fig. 2c, d). The growth defect on raffinose medium is shared between the *brf1*Δ and the *snf5*Δ mutant, but not with the *znf2*Δ mutant. The data suggest that Brf1 has a role in certain stress responses that is independent of Znf2.

Given the striking resemblance of the *brf1*Δ mutant and the *snf5*Δ mutant, we postulated that Brf1 functions in the same biological pathway as Snf5. Indeed, the *brf1*Δ*snf5*Δ double mutant behaved similarly as the *brf1*Δ or the *snf5*Δ single mutant in all

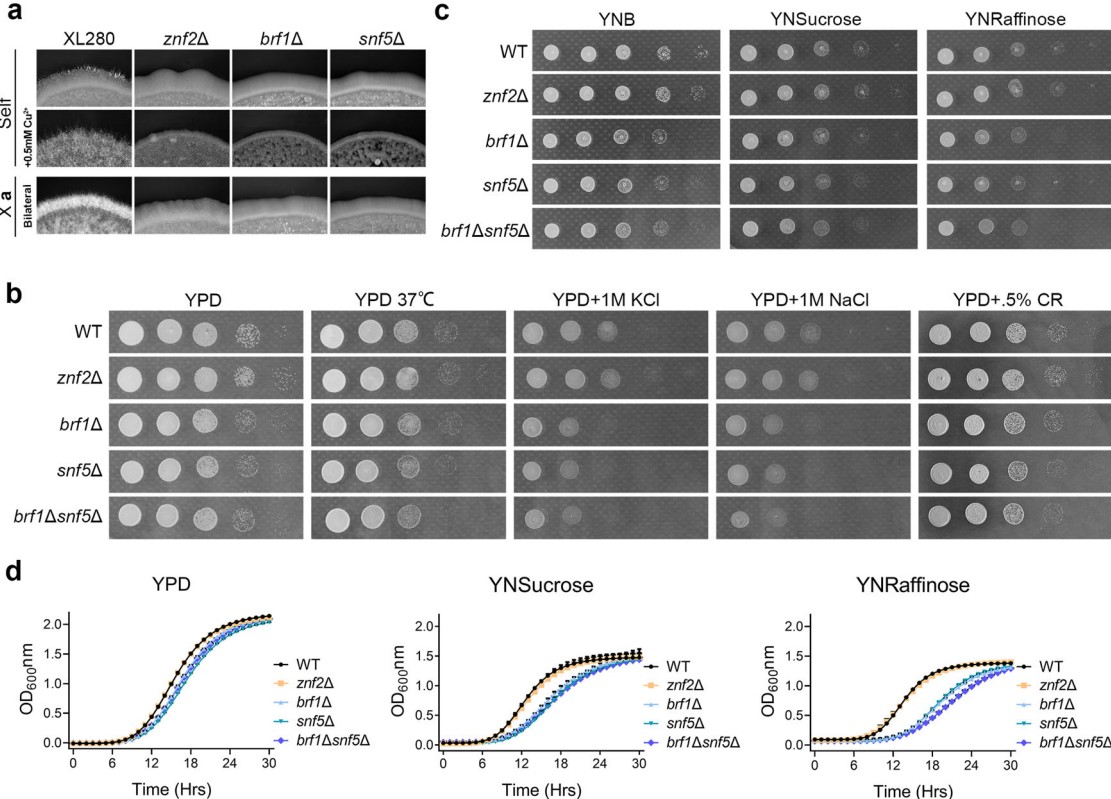

**Fig. 2** The *brf1Δ*, *snf5Δ*, and *brf1Δsnf5Δ* mutants are phenotypically identical. **a** WT XL280, *znf2Δ*, *brf1Δ*, and *snf5Δ* in the α background were cultured on V8 or V8 + 500µM Cu$^{2+}$ medium (self-filamentation), or crossed with the corresponding **a** mating partners (bilateral bisexual mating) on V8 medium in dark for 4 days. **b** WT XL280, *znf2Δ*, *brf1Δ*, *snf5Δ*, and *brf1Δsnf5Δ* strains were cultured on YPD medium at 30 or 37 °C, or with the addition of KCl, NaCl, or Congo Red. **c** WT XL280, *znf2Δ*, *brf1Δ*, *snf5Δ*, and *brf1Δsnf5Δ* strains were cultured on minimal nitrogen base agar media with glucose, sucrose, or raffinose as the sole carbon source. **d** WT XL280, *znf2Δ*, *brf1Δ*, *snf5Δ*, and *brf1Δsnf5Δ* strains were cultured in YPD, YNSucrose, or YNRaffinose broth at 30 °C. The optical density at 600 nm was plotted against the time after inoculation

assays (Fig. 2b–d). Constitutive expression of tdTomato-tagged Brf1 in *brf1Δ* restored filamentation (Supplementary Fig. 4a), indicating functionality of the tagged Brf1. The tagged Brf1 localized to the nucleus (Supplementary Fig. 4a), as reported previously for Snf5 (ref. [33]). Taken together, we propose that Brf1 functions together with Snf5.

As Snf5 is a core subunit of the SWI/SNF complex, we postulate that this novel protein Brf1 may also function in this chromatin remodeling complex. Every known SWI/SNF complex incorporates a subunit with an ARID domain[34–36]. *C. neoformans* carries three genes encoding ARID-containing proteins: *RUM1α* (CND05870), *AVC1* (CNK00710), and *BRF1*. None of these three cryptococcal genes showed high sequence homology to either *SWI1* or *SOL1*, the ARID-containing subunit in the SWI/SNF complex in *S. cerevisiae* or *S. pombe*, respectively. However, among the three proteins, Brf1 is more similar to ScSwi1 or SpSol1 in terms of domain architecture (Fig. 3a). Deletion of *BRF1*, but not *AVC1* or *RUM1α*, caused defects in filamentation (Fig. 3b) and slowed growth on raffinose medium (Supplementary Fig. 3). The result is consistent with the idea that Brf1, but not Avc1 or Rum1α, works in the same complex as Snf5. We, therefore, hypothesize that Brf1 is the ARID subunit of the SWI/SNF complex in this basidiomycete.

**Brf1 is a basidiomycete-specific subunit of SWI/SNF.** If Brf1 and Snf5 work together in the SWI/SNF complex, we expect that Brf1 and Snf5 interact with each other, either directly or indirectly through other subunits of the SWI/SNF complex. To test the

hypothesis, we performed co-immunoprecipitation assays coupled with mass spectrometry using the FLAG-tagged Brf1 as the bait. As *BRF1* is a lowly expressed gene based on our and others' transcriptome data[37], we used the constitutively active *TEF1* promoter to drive the expression of FLAG-tagged Brf1, which restored *brf1Δ*'s mating deficiency (Supplementary Fig. 4b). We then carried out Co-IP/MS in two independent isolates, including the WT H99 strain without any tag as the negative control. In addition to the bait protein Brf1, we identified many proteins homologous to the SWI/SNF subunits in *S. cerevisiae* and/or *S. pombe*, including Snf5, Snf2, Arp4, Arp9, Rsc6, and Rsc8 (Table 1). The interaction between Brf1 and Snf5 was further confirmed by a reciprocal Co-IP/western where the mNeonGreen-tagged Snf5, when used as the bait, pulled down the FLAG-tagged Brf1 (Supplementary Fig. 4c).

The SWI/SNF and the RSC complexes are assembled modularly, with some subunits/modules shared by both complexes[4,38,39]. In *S. cerevisiae* and *S. pombe*, homologs of Arp4, Arp9, Rsc6, and Rsc8 participate in both complexes. Since Brf1 pulled down subunits unique to the SWI/SNF complex and the ones shared by both complexes, we speculate that Brf1 is SWI/SNF-specific in *C. neoformans*. In most fungal species examined, the SWI/SNF complex and the RSC complex each harbors a Snf5 domain-containing subunit, with Snf5 in the SWI/SNF complex and Sfh1 in the RSC complex (Fig. 3c). The mNeonGreen-tagged RSC-specific protein Sfh1 (CNC06140) in *C. neoformans* pulled down RSC-specific subunits including Rsc1, Rsc7, and Rsc9. Sfh1 also pulled down the shared subunits including Snf2, Arp4, Arp9, Rsc6, and Rsc8 (Supplementary Table 2). However, Sfh1 did not

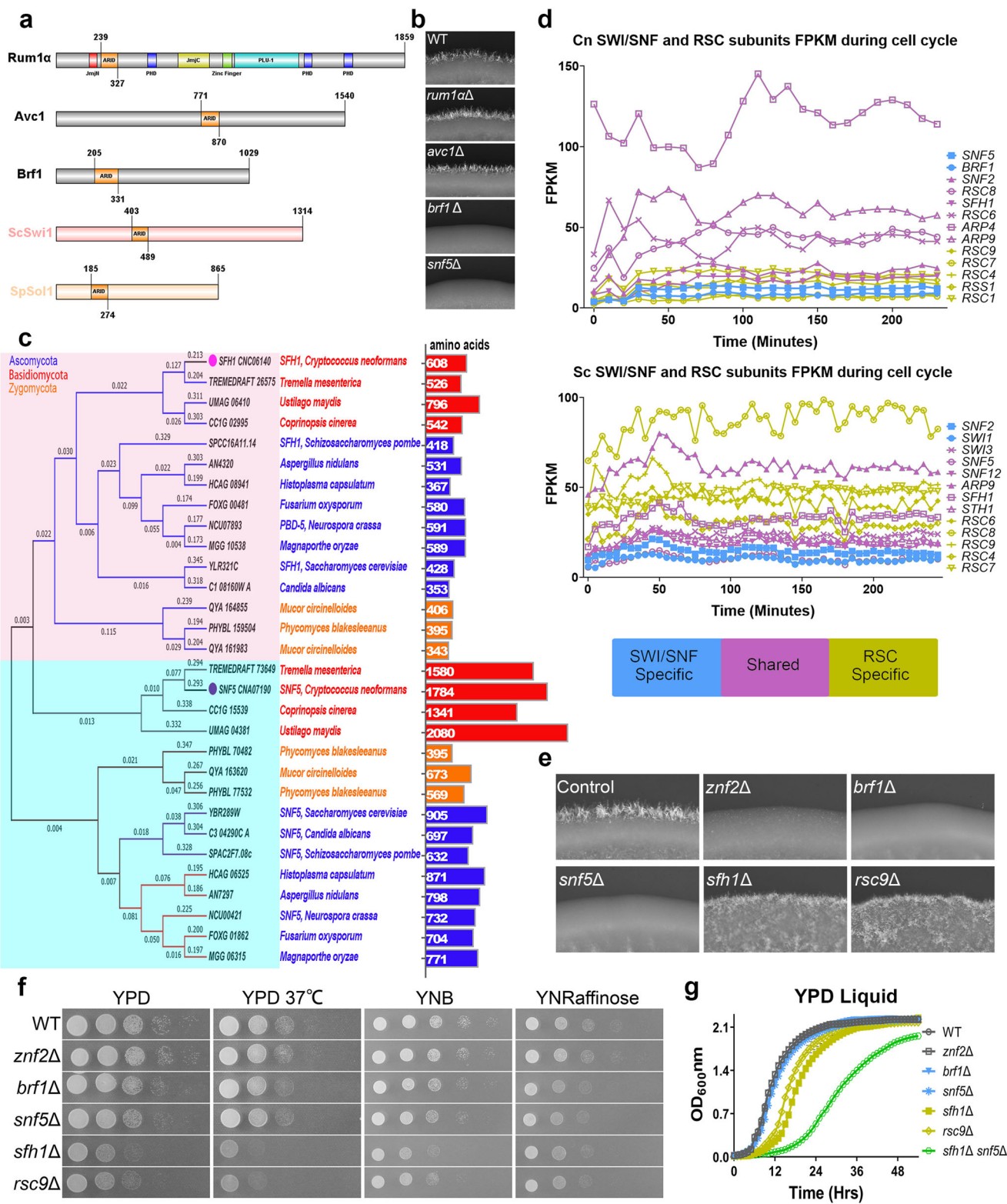

pull down Brf1 or the SWI /SNF-specific subunit Snf5. Collectively, these lines of evidence support that Brf1 is a subunit specific to the SWI/SNF complex.

**SWI/SNF but not RSC is critical for hyphal differentiation.** Based on the aforementioned evidence, we inferred that Brf1 and Snf5 are subunits specific to the SWI/SNF complex while Sfh1,

Rsc1, Rsc4, Rsc7, and Rsc9 are subunits specific to the RSC complex in *C. neoformans* (Table 2). Snf2, Arp4, Arp9, Rsc6, and Rsc8 are the shared subunits (Table 2, Supplementary Fig. 5a). Consistent with the idea that these two complexes have overlapping and distinct subunits, the transcript levels of the shared components are generally higher (purple) than the complex-specific subunits (yellow or blue) based on a cell-cycle-regulated transcriptome data[37] (Fig. 3d). In both *S. cerevisiae* and *S. pombe*,

**Fig. 3** Brf1 and Snf5 are both subunits specific to the SWI/SNF complex. **a** Diagrams of domain organization of the three ARID-containing proteins in *C. neoformans* (Rum1α, Avc1, and Brf1) and the SWI/SNF ARID-containing subunit Swi1 in *S. cerevisiae* and Sol1 in *S. pombe*. **b** WT XL280, *snf5Δ*, *rum1αΔ*, *avc1Δ*, and *brf1Δ* strains were cultured on V8 agar medium at 22 °C in dark for 2 days. **c** A phylogenetic tree of Snf5 domain-containing proteins in the selected ascomycetes, basidiomycetes, and zygomycetes. The number of amino acids for each protein was indicated on the right. **d** The FPKM values for genes encoding the selected SWI/SNF and RSC subunits in *C. neoformans* and *S. cerevisiae* were plotted against the time across the cell cycle. The SWI/SNF-specific subunits are in blue, the RSC-specific subunits are in yellow, and the shared subunits are in purple. **e** WT XL280, *znf2Δ*, *brf1Δ*, *snf5Δ*, *sfh1Δ*, and *rsc9Δ* strains were cultured on V8 agar medium at 22 °C in dark for 2 days. **f** WT XL280, *znf2Δ*, *brf1Δ*, *snf5Δ*, *sfh1Δ*, and *rsc9Δ* strains were cultured on YPD agar medium at 30 °C or at 37 °C, YNB or YNRaffinose agar medium at 30 °C. **g** WT XL280, *znf2Δ*, *brf1Δ*, *snf5Δ*, *sfh1Δ*, *rsc9Δ*, and *snf5Δsfh1Δ* strains were cultured in YPD broth. The optical density at $OD_{600}$ nm was plotted against the time after inoculation. The SWI/SNF-specific subunits are in blue and the RSC-specific subunits are in yellow

---

**Table 1 The list of proteins identified from Co-IP/MS by Brf1-CBP-2×FLAG as bait**

| Coding locus (D) | Coding locus (A) | Protein name | JL401 (strain 1) | JL402 (strain 2) |
|---|---|---|---|---|
| | | | # peptide spectrum matches | |
| CNE04020 | CNAG_02134 | Rsc8[a] | 12 | 20 |
| CNK02030 | CNAG_01863 | Snf2[a] | 10 | 23 |
| CNI00980 | CNAG_04460 | Arp9[a] | 8 | 9 |
| CNA07190 | CNAG_00740 | Snf5[b] | 4 | 5 |
| CND01230 | CNAG_00995 | Msc1 | 4 | 3 |
| CNG02900 | CNAG_03285 | Rsc6[a] | 3 | 10 |
| CNB05320 | CNAG_04048 | Arp4[a] | 3 | 4 |
| CNE02000 | CNAG_02350 | Rss1[a] | 3 | 3 |
| CNA02310 | CNAG_00240 | Brf1[b] | 2 | 6 |
| CNA00820 | CNAG_00091 | N/A | 2 | 2 |
| CNK02620 | CNAG_01920 | Ubi4 | 2 | 2 |

[a]The SWI/SNF and RSC complex shared subunits [b]The SWI/SNF-specific subunits

---

**Table 2 The BLAST analysis of the SWI/SNF and RSC subunits in *C. neoformans*, *S. cerevisiae* (selected subunits), and *S. pombe* (selected subunits)**

| *C. neoformans* | | *S. pombe* | | *S. cerevisiae* | |
|---|---|---|---|---|---|
| Subunit | Deletion phenotype | Subunit | Deletion phenotype | Subunit | Deletion phenotype |
| Snf5[a] | viable, non-filamentous | Snf5[a] | Viable | Snf5[a] | Viable |
| | | Sol1[a] | Viable | Swi1[a] | Lethal/Viable[b] |
| Brf1[a] | Viable, non-filamentous | | | | |
| Snf2[c] | Viable, reduced filament | Snf22[a] | Viable | Snf2[a] | Viable |
| Rsc8[c] | | Ssr1[c] | Lethal | Swi3[a] | Viable |
| | | Ssr2[c] | Lethal | | |
| Rsc6[c] | | Ssr3[c] | Lethal/viable[b] | Snf12[a] | Viable |
| | | Ssr4[c] | Lethal/viable[b] | | |
| Arp9[c] | | Arp9[c] | Viable | Arp9[c] | Lethal or sick |
| | | | | Arp7[c] | Lethal or sick |
| Arp4[c] | | Arp42[c] | Viable | | |
| Rss1[c] | | | | | |
| | | | | Rsc8[d] | Lethal |
| | | | | Rsc6[d] | Lethal |
| Snf2[c] | Viable, reduced filament | Snf21[d] | lethal | Sth1[d] | lethal |
| Sfh1[d] | Viable, filamentous | Sfh1[d] | Lethal | Sfh1[d] | Lethal |
| Rsc9[d] | Viable, filamentous | Rsc9[d] | Lethal | Rsc9[d] | Lethal |
| Rsc7[d] | | Rsc7[d] | Lethal | Rsc7[d] | Viable |
| Rsc1[d] | | Rsc1[d] | Viable | Rsc1[d] | Double mutant lethal |
| | | | | Rsc2[d] | |
| Rsc4[d] | | Rsc4[d] | Viable | Rsc4[d] | Lethal |

[a]The SWI/SNF-specific subunits [b]The phenotype depends on strain backgrounds [c]the SWI/SNF and RSC complex shared subunits [d]The RSC-specific subunits

---

many subunits in the RSC complex are essential for growth, including Sfh1 and Rsc9 (Table 2)[5,40,41]. By contrast, the RSC-specific *SFH1* and *RSC9* genes are not essential for growth in *C. neoformans*, although the *sfh1Δ* and the *rsc9Δ* mutants grew slower (Fig. 3f, g). Surprisingly, deletion of the SNF/SWI complex-specific *BRF1* or *SNF5* did not cause any growth defects under the same conditions (Fig. 3f, g). Remarkably, the *snf5Δsfh1Δ* double mutant, where both the SWI/SNF-specific subunit Snf5 and the RSC-specific subunit Sfh1 were disrupted, was still viable despite a much more pronounced growth defect (Fig. 3g). Thus, the SWI/SNF complex functionally differ from the RSC complex in *C. neoformans*. Neither the RSC complex nor both RSC and SWI/SNF complexes together are essential in this basidiomycetous fungus, in contrast to what is known in ascomycetes.

Given that the two SWI/SNF complex-specific subunits Brf1 and Snf5, and none of the RSC complex-specific subunits were identified from our genetic screen, we predict that the SWI/SNF complex, but not the RSC complex, specifically regulates the yeast-hypha differentiation in *C. neoformans*. To test this hypothesis, we examined the impact of the disruption of the SWI/SNF complex (*brf1*Δ and *snf5*Δ) or the RSC complex (*sfh1*Δ and *rsc9*Δ) on filamentation. Indeed, the RSC complex mutants, *sfh1*Δ and *rsc9*Δ, were still filamentous on V8 medium, in contrast to the yeast growth of the SWI/SNF complex mutants *brf1*Δ and *snf5*Δ (Fig. 3e).

**BRF1 is required for the induction of the ZNF2 transcripts.** As Brf1 and Snf5 work together in the SWI/SNF complex, we decided to use Brf1 to further dissect the relationship between the SWI/SNF complex and Znf2 in controlling morphogenesis in *C. neoformans*. To study the genetic relationship between *BRF1* and *ZNF2*, we crossed a *brf1*Δ/*BRF1*[oe] α strain with a *znf2*Δ/*ZNF2*[oe] **a** strain. We micro-dissected meiotic progeny from the cross and confirmed the genotypes of the progeny by diagnostic PCRs (Fig. 4a). As expected, the progeny of the wild-type genotype were self-filamentous, while the progeny of the *znf2*Δ or the *brf1*Δ genotype were non-filamentous. *ZNF2*[oe]*znf2*Δ strains were filamentous, but not *ZNF2*[oe]*brf1*Δ (Fig. 4b). This result is consistent with our earlier observation that overexpression of *ZNF2* did not restore filamentation in the *brf1*[Tn] insertional mutant (Fig. 1c).

Likewise, *BRF1*[oe]*brf1*Δ strains were filamentous, but not *BRF1*[oe]*znf2*Δ (Fig. 4b). Thus, overexpression of either *BRF1* or *ZNF2* cannot override the absence of the other, and both are essential for yeast-hypha differentiation in *C. neoformans*.

We next analyzed the transcriptomes of WT, *brf1*Δ, *brf1*Δ/ *BRF1*[oe], and *ZNF2*[oe] strains cultured under filamentation-repressing YPD medium and filamentation-inducing V8 medium (Supplementary Data 2). More than 60% of the differentially expressed genes (up- or down-regulated) in *brf1*Δ were also differentially expressed in the *znf2*Δ mutant under filamentation-inducing condition (Fig. 4c). Remarkably, >95% (95/99) of the overlapping genes were down-regulated upon deletion of *BRF1* or *ZNF2*, suggesting that Brf1 and Znf2 activate these genes during hyphal differentiation. As *ZNF2*/filamentation is primarily induced by the pheromone pathway when cells are cultured on V8 medium[19], it is not surprising that deletion of either *BRF1* or *ZNF2* dampened the induction of the pheromone pathway genes: pheromone *MF*α (181× reduction), pheromone receptor *STE3*α (14× reduction), pheromone exporter *STE6* (7× reduction), MAPK *CPK1* (5.1× reduction), and pheromone transcription factor *MAT2* (9× reduction) (Fig. 4c, right panel). The reduced activation of the pheromone pathway in the *brf1*Δ mutant lowered cell fusion which we observed earlier (Supplementary Fig. 2b). On the other hand, much fewer genes were differentially expressed in *brf1*Δ and *znf2*Δ under filamentation-suppression YPD condition (Supplementary Fig. 6, Supplementary Data 2).

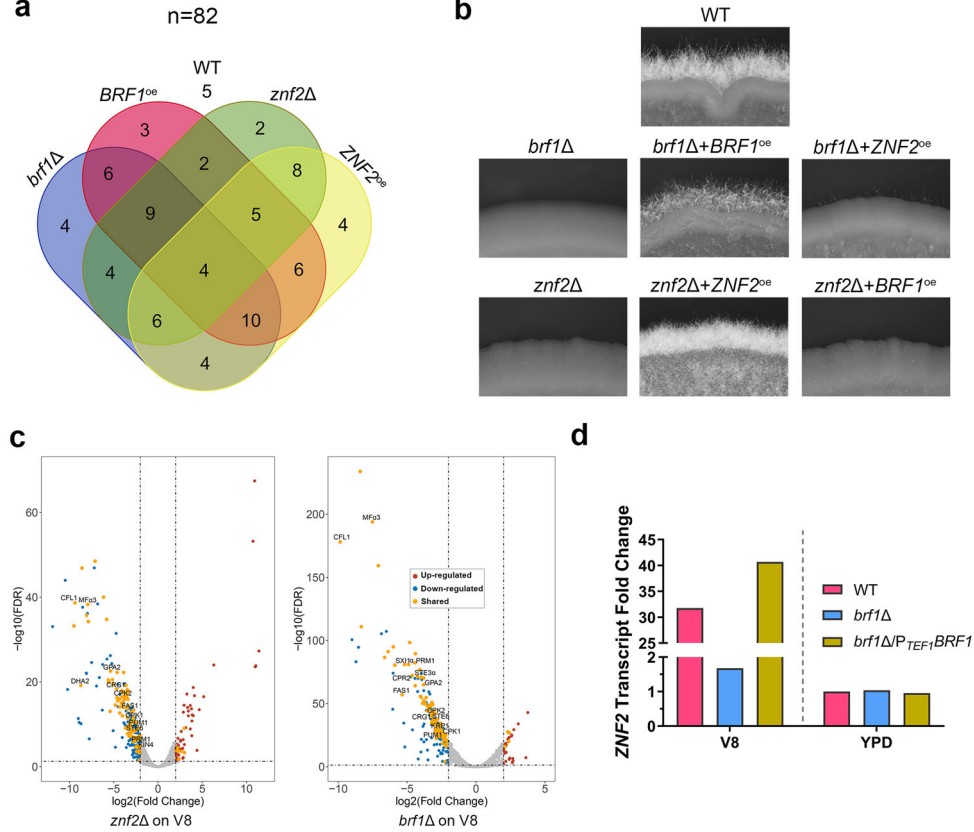

**Fig. 4** *BRF1* is required for *ZNF2* transcription induction during hyphal differentiation. **a** The Venn diagram showing the number of progeny for each genotype from a cross between *znf2*Δ/*ZNF2*[oe] **a** and *brf1*Δ/*BRF1*[oe] α . *n* indicates the total meiotic spores analyzed. **b** WT XL280 and selected progeny of the following genotypes (*brf1*Δ, *brf1*Δ + *BRF1*[oe], *brf1*Δ + *ZNF2*[oe], *znf2*Δ, *znf2*Δ + *ZNF2*[oe], and *znf2*Δ + *BRF1*[oe]) were cultured on V8 + BCS medium (to induce *ZNF2*) at 22 °C in dark for 5 days. **c** Volcano plots of fold changes in transcript level in the *znf2*Δ mutant and the *brf1*Δ mutant compared to WT cultured on V8 agar medium for 24 h. Each dot in the plots indicates a protein-coding gene. The vertical dash lines indicate the $|\log_2^{FC}| = 2$ and the horizontal dash line shows the FDR = 0.05. The differentially up-regulated genes are indicated in red and the differentially down-regulated genes are indicated in blue. The shared DEGs between the *znf2*Δ and the *brf1*Δ mutants are colored in orange. **d** The relative transcript levels of *ZNF2* in WT, *brf1*Δ, and *brf1*Δ/*BRF1*[oe] cultured on V8 agar medium and YPD medium. The *ZNF2* transcript level in WT cultured on YPD medium was used for normalization and was set as 1

The transcript level of *ZNF2* in WT increased 32-fold under filamentation-inducing condition compared to filamentation-suppressing conditions. In the absence of *BRF1*, however, the *ZNF2* transcript level remained at the basal level as observed under filamentation-suppressing conditions (Fig. 4d). Ectopic overexpression of *BRF1* in the *brf1Δ* strain restored the dramatic induction of the *ZNF2* transcript level under filamentation-inducing conditions (Fig. 4d). By contrast, constitutive expression of *ZNF2* (54.75× increase) had minimal impact on the transcript level of *BRF1* (1.27× increase). The transcript of *BRF1* remained at a low and steady level when cells were cultured either in YPD or on V8 medium. The steady level of *BRF1* transcripts was also observed at different cell cycle stages (Fig. 3d)[37]. The results indicate that *BRF1* is expressed at a low but constant level and is not influenced by Znf2. The induction of *ZNF2*, however, requires Brf1.

**Brf1 is required for Znf2's full association to its targets**. Brf1 is required for *ZNF2* induction; however, even when *ZNF2* was ectopically overexpressed, Znf2 still failed to drive filamentation in the absence of Brf1 (Figs. 1c and 4b). Therefore, beyond transcription activation of *ZNF2*, Brf1 is critical for existent Znf2 protein to execute its function. We hypothesize that Znf2's association with DNA may require Brf1.

To determine if the ability of Znf2 to bind to its downstream targets is affected by the SWI/SNF complex, we conducted chromatin immunoprecipitation (ChIP) experiments using 3xFLAG-tagged Znf2 as the bait in the WT or the *brf1Δ* mutant background. We first used qPCR to compare the relative abundance of the precipitated genetic locus of the filamentation marker gene *CFL1*, a proxy for Znf2 downstream targets[20,24,42,43]. We found that Znf2 strongly binds to the *CFL1* promoter (Supplementary Fig. 7a). We also found that Znf2 strongly binds to its own promoter (Supplementary Fig. 7b), suggesting that Znf2 likely autoregulates itself. In the absence of Brf1, the association of Znf2 to the promoter regions of *CFL1* and *ZNF2* decreased 2–3-fold (Fig. 5a, Supplementary Fig. 7). Thus, the SWI/SNF complex helps Znf2 associate with its downstream targets, including *ZNF2* itself.

We further investigated the requirement of Brf1 for Znf2's binding to its targets by analyzing the sequencing results of the same ChIP samples. A total of 361 potential Znf2-binding regions were identified (Fig. 5b, Supplementary Data 3). Consistent with our ChIP-qPCR results (Fig. 5a), Znf2's binding to most of its downstream sites, including the *ZNF2* and *CFL1* promoter regions, was dampened upon the disruption of Brf1 (Fig. 5b, f). The finding is consistent with our earlier transcriptome data showing that *BRF1* is required for *ZNF2* transcription induction and that *brf1Δ* shares more than 60% of the differentially expressed genes with *znf2Δ* when cultured under filamentation-inducing conditions (Fig. 4d). Thus, Brf1, a subunit of the chromatin-remodeling SWI/SNF complex, is required for full association of the transcription factor Znf2 with its downstream targets.

**Brf1 and Znf2 work in concert in cellular differentiation**. The SWI/SNF complex is known to evict or slide nucleosomes to change chromatin structure. To test if Brf1 affects cell differentiation through chromatin remodeling of *ZNF2* and Znf2 target regions, we employed the Assay for Transposase-Accessible Chromatin using sequencing (ATAC-seq)[44,45] and compared genetic regions with open chromatin structures in the WT strain, grown under filamentation-repressing or -inducing conditions, as well as in the *brf1Δ*, *znf2Δ*, and *snf5Δ* strains cultured under filamentation-inducing conditions. We included the *brf1Δ/*

*BRF1*[oe] strain as an additional control. As expected, accessible chromatin fell in the promoter regions, with expressed genes having a higher level of accessibility based on the relative enrichment of ATAC-seq reads (Fig. 5c). Globally, chromatin accessibility was similar in all strains tested (Fig. 5d; Supplementary Data 4). When we examined the chromatin accessibility of the 361 regions that Znf2 potentially binds to based on the ChIP data (Fig. 5b; Supplementary Data 3), we noticed a significant reduction in chromatin accessibility in *brf1Δ* and *snf5Δ* across most of the regions, which now resembled the accessibility pattern for WT grown under the filamentation-suppression condition (Fig. 5e). K-means clustering further showed that Znf2-binding sites in clusters 1 and 2 (83 regions) did not strongly depend on Brf1, Snf5, or Znf2 to maintain open chromatin. Cluster 3 peaks (118 Znf2-binding regions) had low accessibility in YPD but high accessibility on V8. Accessibility of cluster 3 peaks was largely dependent on Brf1, Snf5, and Znf2. Cluster 5 (138 peaks) showed moderate accessibility on V8, which depended on Brf1, Snf5, and Znf2. Cluster 4 (22 peaks) was highly accessible on V8 and this depended on Brf1 and Snf5, but less dependent on Znf2. Fifteen out of the 19 genes potentially affected by the Brf1-dependent ATAC-seq peaks (cluster 4) overlapped with genes up-regulated under hyphal-promoting conditions based on the transcriptome data (Supplementary Data 3, colored in red). The genomic coordinates, cluster number, and nearest gene promoter for each Znf2-binding region are listed in Supplementary Data 3.

We next identified regions with differential ATAC enrichment in WT and *brf1Δ* cultured in filamentation-inducing conditions (Supplementary Data 5). Forty-one accessible regions were dependent on Brf1 (Supplementary Data 5, labeled in red), including regions immediately upstream of *CFL1* and *ZNF2*. Surprisingly, 1272 regions were identified as hyper-accessible in *brf1Δ* compared to WT. A heatmap to visualize ATAC-reads across all differential peaks revealed that hyper-accessible regions identified in *brf1Δ* were associated with reduced accessibility at adjacent sites (Supplementary Fig. 8). Thus, these hyper-accessible regions may accumulate as a result of reduced nucleosome mobility at these sites in *brf1Δ*. Ninety-four percent of differential peaks in *brf1Δ* overlapped a differential peak identified in *snf5Δ*, consistent with Brf1 and Snf5 functioning together in the same complex (Fig. 5d, Supplementary Data 6).

RNA-seq, ChIP-seq, and ATAC-seq reads for three of these genes, along with a control gene CNA07690 that does not require Brf1 for its open chromatin are shown in Fig. 5f. Most noticeably, the promoter region of the filamentation marker gene *CFL1* was the top differentially accessible region (10.02× change) (Fig. 5d, Supplementary Data 5). *CFL1* is one of the highest induced genes controlled by Znf2 (ref. [21]). Disruption of either Brf1, Snf5, or Znf2 abolished accessibility of the *CFL1* promoter (ATAC-seq), where Znf2 binds to (ChIP-seq) (Fig. 5b, f). Accordingly, the *CFL1* transcript level was almost undetectably low when *BRF1* or *ZNF2* was deleted. Overexpression of *BRF1* in *brf1Δ* restored chromatin accessibility of the *CFL1* promoter and also its transcript level (Fig. 5f). Similarly, we found that the *ZNF2* promoter region became inaccessible in the *brf1Δ* and the *znf2Δ* mutants (5.88-fold change; Supplementary Data 5).

Collectively, the data support the interdependence between the transcription factor Znf2 and the SWI/SNF complex in opening up chromatin to facilitate transcription of filamentation genes.

## Discussion
Chromatin remodeling plays critical roles in cellular differentiation in eukaryotes. The model ascomycetous yeasts *S. cerevisiae* and *S. pombe* have offered a relatively simple system for mechanistic

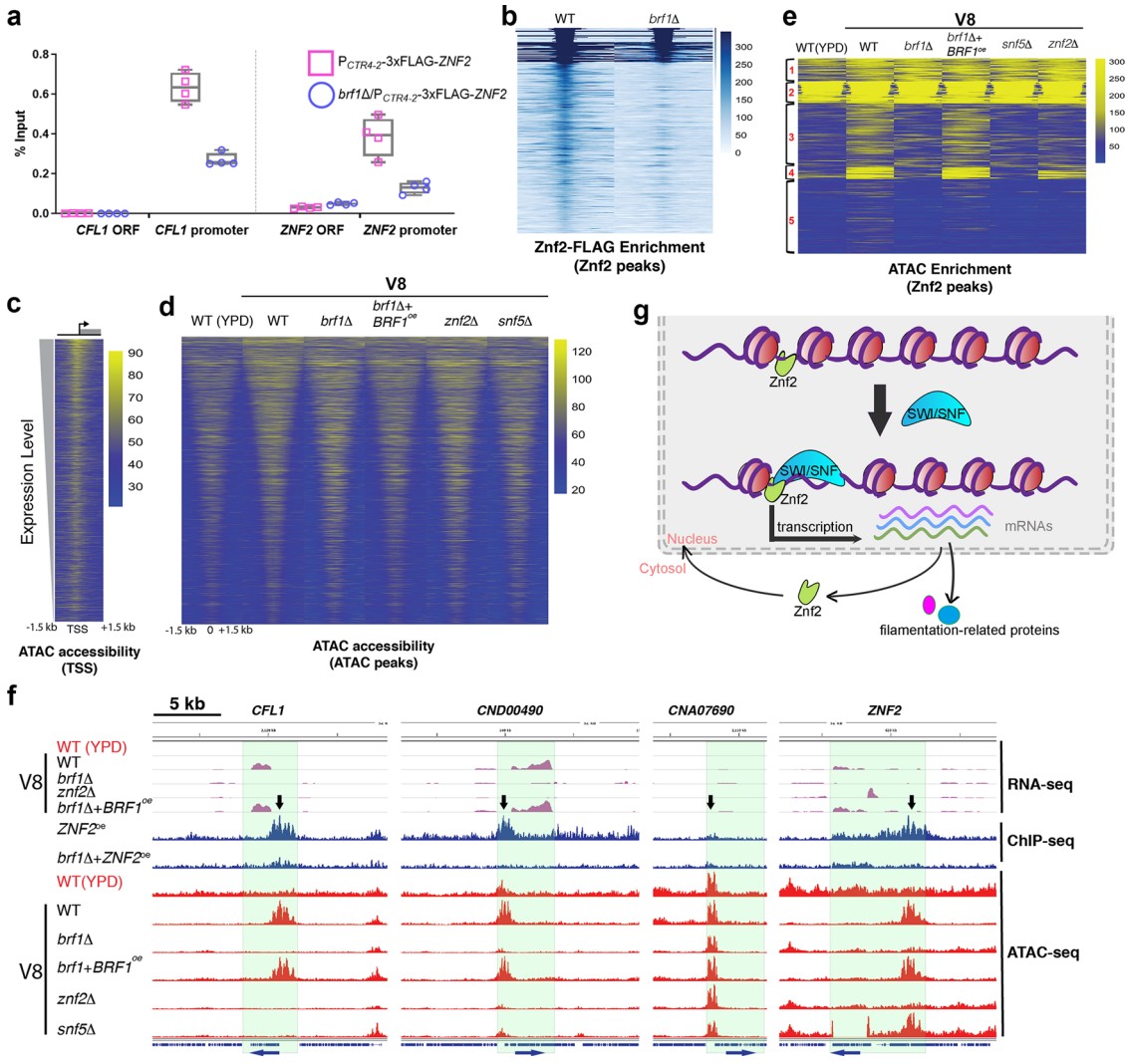

**Fig. 5** Znf2 and the SWI/SNF complex coordinate in transcription activation of filamentation genes. **a** The relative abundance of DNAs associated with Znf2 in designated strains grown in YPD + BCS media based on ChIP-qPCR. The points represent data from two technical replicates of each biological duplicate. **b** The heatmap of the relative enrichment of Znf2-FLAG binding regions in WT and the *brf1Δ* mutant across all potential 361 Znf2-binding sites identified from ChIP-seq (lowest *p* value on top and the highest *p* value on bottom). **c** The heatmap depicts relative enrichment of ATAC-seq reads from WT grown on V8 medium. The peaks are centered on the transcription start sites (TSS) for all genes arranged from the highest expression level (top) to the lowest expression level (bottom). **d** Global chromatin accessibility in the tested mutants. The relative enrichment of ATAC-seq reads for all ATAC-seq peaks from the WT strain grown on V8 medium is shown for each indicated strain. **e** Relative enrichment of the ATAC-seq peaks over the 361 Znf2-FLAG binding regions in WT, *brf1Δ*, *brf1Δ + BRF1^oe*, *snf5Δ*, and *znf2Δ* cells. **f** Genome browser images depict relative transcript levels based on RNA-seq (purple), ChIP-seq (blue), and ATAC-seq enrichment (red) for the four indicated genetic regions of the indicated strains. Images depicting representative ATAC-seq peaks at the three genetic loci that display reduced ATAC-seq enrichment in *brf1Δ* (*CFL1*, *CND00490*, and *ZNF2*) as well as one control genetic locus (*CNA07690*) that does not exhibit altered accessibility with or without Brf1. WT grown in YPD or on V8 medium is used as the negative or the positive control for filamentation. The down arrows indicate the location of differential accessible regions. The arrows at the bottom of the plot indicate direction of transcription. **g**. A diagram of the working model depicting the coordination between Znf2 and the SWI/SNF complex in regulating gene transcription. Znf2 binds to its target sites at a low basal level. Upon culturing under filamentation-inducing condition, Znf2 recruits the SWI/SNF complex to its target sites to open chromatins and activate transcription of genes important for filamentation

studies. Basidiomycetous fungi resemble more of the higher eukaryotes in terms of genome structures, epigenetic regulation, and transcriptome complexity, but research on chromatin remodeling in this major phylum of the fungal kingdom is scarce.

To our knowledge, our study is the first to identify and characterize a basidiomycete-specific factor that serves as a critical subunit in the SWI/SNF complex. We demonstrated that this phylum-specific factor Brf1, together with the conserved known subunit Snf5 of the SWI/SNF complex, are essential for hyphal growth and sexual development in the basidiomycete *C. neoformans*. This complex remodels the chromatin structure of the

promoter regions of filamentation genes to make them accessible for transcription (Fig. 5g). Brf1 is vital for transcriptional induction of *ZNF2* and also full association of Znf2 protein to the promoters of its downstream targets (e.g. *CFL1* and *ZNF2*). Our findings are consistent with published literature in other organisms, in which the SWI/SNF complex contributes to the DNA binding of the transcription factor[46,47]. The SWI/SNF complex has been shown being targeted to specific genetic loci by sequence-specific transcription factors and acetylation of histone tails[46,48,49]. In the ascomycetous fungus *C. albicans*, the transcription factor Efg1 recruits the histone acetyltransferase

complex NuA4 to the promoters of hypha-associated genes, allowing, in turn, the recruitment of the SWI/SNF complex to activate their transcription[50]. We reason that Znf2 may recruit the SWI/SNF complex to its target sites given that accessibility at the *CFL1* promoter region is lost in the absence of *ZNF2* (Fig. 5b–f). Although we could not establish direct physical interaction between Znf2 and the SWI/SNF complex via Co-IP experiment, the interaction may be too weak or transient to be captured by our assay. Alternatively, histone modifications or the modification enzymes may bridge the connection. Several SWI/SNF complex subunits possess domains that recognize histone modifications, such as bromodomains for histone acetylation[51]. How exactly the SWI/SNF complex works with Znf2 in *C. neoformans* to control differentiation warrants further investigation.

Our findings revealed striking differences in the two complexes (SWI/SNF and RSC) between Ascomycota and Basidiomycota. In *Saccharomyces* and *Schizosaccharomyces* species, most subunits in the RSC complex are essential for growth, including the catalytic ATPase subunit Snf2 (or SpSnf22), Sfh1, and Rsc9 (refs. [5,40,41]) (Table 2). Snf2 is not essential in *C. neoformans*. Consistently, deletion of both *SNF5* (SWI/SNF-specific) and *SFH1* (RSC-specific) together slowed growth but did not cause lethality (Fig. 3g). Comparative functional analyses of the RSC complex among evolutionarily diverse species, therefore, provides a unique vantage point to understand the biological function of the RSC complex.

All SWI/SNF complexes have a subunit with a conserved Snf5 domain, which acts as a scaffold protein. The majority of fungi examined carry two proteins bearing a Snf5 domain (Pfam domain ID PF04855), presumably one acting as the Snf5 subunit in the SWI/SNF complex and one acting as the Sfh1 subunit in the RSC complex (Fig. 3c, Supplementary Table 3). Based on primary protein sequences, Snf5 proteins in Ascomycota separate into two clades, Snf5 and Sfh1 (Fig. 3c). *C. neoformans* and other basidiomycetes harbor two proteins with an Snf5 domain: *SNF5* and *SFH1* and they cluster closely with the Sfh1 clade in ascomycetes (Fig. 3c). The SWI/SNF complex assembles in an ordered modular fashion[4,38,39]. In *Saccharomyces*, Snf5 and Swi1 (the ARID-containing protein) likely belong to two different modules[4,38,39] given the phenotypical differences of the *snf5* and *swi1* mutants[52]. However, *Cryptococcus snf5* and *brf1* mutants have nearly identical phenotypes. Furthermore, Snf5 proteins in basidiomycetes (*C. neoformans*: 1784 aa, *U. maydis*: 2080 aa) are much larger in size compared to those in ascomycetes or in humans (*S. cerevisiae*: 905 aa, *S. pombe*: 632aa, or humans: 385aa) (Fig. 3c). Two internal repeats near the N-terminus of Snf5 proteins in basidiomycetes (Supplementary Fig. 5b) could potentially increase binding surface area and assist assembling the SWI/SNF complex.

The SWI/SNF family employs Snf2 as the ATPase catalytic subunit, which carries an HSA and a BROMO domain (the DEXDc and HELICc regions)[3]. A search for the HSA domain (PF07529) and the BROMO domain (PF00439) revealed that some fungi, including ascomycete *S. cerevisiae* and zygomycete *Mucor circinelloides*, have two distinct Snf2 family ATPases, presumably one for the SWI/SNF complex and one for the RSC complex. Some other fungi, including ascomycete *Aspergillus nidulans* and basidiomycete *C. neoformans*, however, have only one Snf2 ATPase, presumably shared by SWI/SNF and RSC (Table 2, Supplementary Table 4). In *Drosophila melanogaster*, the BAP (SNF/SWI) complex and the PBAP (RSC) complex share the same catalytic ATPase subunit (BRM). Human BAF (mSNF/SWI) complex and PBAF (mRSC) complex can share the same ATPase BRG-1 subunit or use different ATPases (hBRM and BRG-1)[3]. Thus, the ATPase could be shared by or be unique to the SWI/SNF and the RSC complexes. The copy number variation of *SNF2*

might have driven such divergence independent of the evolutionary distance of the species.

In conclusion, we discovered the basidiomycete-specific subunit of SWI/SNF chromatin remodeling complex and revealed the major differences in the composition and biology of this complex between ascomycetes and basidiomycetes. Given the universal importance of the SWI/SNF complex in cellular differentiation in eukaryotes, our findings provide an important platform for future comparative functional analyses of this complex in diverse eukaryotic lineages.

## Methods and materials

**Strains and growth conditions**. Strains used in this study are listed in Supplementary Table 5. All strains were stored in 15% glycerol at −80 °C and were freshly streaked out onto YPD media (2% peptone, 1% yeast extract, 2% glucose, 2% agar, grams/liter) for each experiment. *C. neoformans* cells were cultured in YPD medium unless stated otherwise. For some assays, the defined minimal YNB medium (6.7 g/L nitrogen base w/o amino acids, 2% agar, 2% glucose) was used as specified in the figure legends. For mating or filamentation assays, V8 juice agar medium (1 liter medium, 50 mL V8 original juice, 0.5g KH$_2$PO$_4$, 4% agar, pH 5.0 or 7.0) was used.

***Agrobacterium*-mediated T-DNA insertional mutagenesis**. The *Agrobacterium*-mediated cryptococcal transformation was carried out as described previously[53]. Briefly, engineered *Agrobacterium* cells that carry a Ti plasmid with the hygromycin-resistant marker were co-incubated with the recipient *Cryptococcus* cells (P$_{CTR4-2}$-mCherry-*ZNF2*-*NEO*, *znf2*::*NAT*) on the induction medium at 22 °C for 2 days. The *Cryptococcus*–*Agrobacterium* cocultures were collected, diluted, and transferred onto the selection medium (with 50 μM of BCS) that would induce *ZNF2* expression. Cryptococcal transformants were cultured on this selective medium for 3–5 days at 22 °C in dark. Approximately 500 colonies grew on each selective plate and about 88,000 transformants were screened for smooth yeast colonies.

**Insertion site identification in the selected AMT mutants**. Genomic DNA of the selected mutants was prepared with the CTAB DNA extraction protocol as described previously[54]. The genomic DNA for each strain was then pooled together with equal molar concentration and the pooled DNA was sent for sequencing (Illumina MiSeq, 250 bp × 250 bp, paired-end reads, BioProject accession number: PRJNA534125, SRA: SRR8943502). The insertion sites in the mutants were identified via the AIMHII program developed by Esher et al.[26] with default parameters.

**Gene deletion, gene overexpression, and protein tagging**. The gene knockout constructs were constructed by fusing 5′ and 3′ homologous arms (each about 1 kb) with dominant drug marker via overlap PCRs. The constructs were introduced into cryptococcal cells by biolistic transformation or by TRACE (Transient CRISPR-Cas9 Coupled with Electroporation) as described previously[55,56]. Single-guide RNAs were designed with Eukaryotic Pathogen CRISPR guide RNA/DNA Design Tool (http://grna.ctegd.uga.edu/) for TRACE.

To generate gene overexpression strains, the ORF amplicon with added *Fse*I and *Asi*SI cutting sites was digested and ligated into the vectors where the ORF was placed downstream of the *GPD1*, the *TEF1*, or the *CTR4-2* promoter, as we described previously[42]. A fluorescent tag (tdTomato or mNeonGreen) or a CBP (calmodulin-binding peptide)-2×FLAG tag was placed immediately in-frame downstream of the *Asi*SI cutting site, which allows in frame tagging of the protein at the C-terminus.

*ZNF2* ORF was ligated into a vector where a 3xFLAG tag was downstream of the *CTR4* promoter to allow tagging at the N-terminus. The functionality of the tagged proteins was confirmed by their ability to rescue/restore the phenotypes in the corresponding gene deletion strains. All constructed plasmids used in this study are listed in Supplementary Table 6. All the primers used for constructing or confirming gene deletion, gene overexpression, or protein tagging are listed in Supplementary Table 7.

**Phenotypic assays.** For phenotypic analyses, cells of the tested strains were cultured in YPD broth overnight at 30 °C with shaking at 230 r.p.m. Cells were collected by centrifugation, washed twice with sterile water, and then suspended with water to the same optical density at 600 nm (OD$_{600}$ = 3). Three microliters of cell suspension and 10× serial dilutions were spotted onto YPD agar medium or YPD with the supplement of Congo Red (0.3–0.5%) and were incubated at 30 or 37 °C for 1–2 days. To test carbon source utilization, glucose in the YNB medium was replaced by either sucrose or raffinose (final concentration: 2%).

To test the ability of these strains to undergo self-filamentation, 3 μL of cells with OD$_{600}$ = 3 were spotted onto V8 or V8 + 500 μM Cu$^{2+}$ medium and incubated at 22 °C in dark for 2–10 days. To examine filamentation during bisexual mating, an equal number of cells of compatible mating types were mixed, spotted onto V8 agar medium (pH 5 for mating assay of serotype A strains and pH 7 for mating assay of serotype D strains) and incubated at 22 °C in dark for 1–7 days.

For strains that use the *CTR4-2*-inducible promoter to drive the expression of the examined genes, cells were maintained on YPD medium with 50 μM CuSO$_4$ to suppress the gene expression. To induce the gene expression, the copper chelator BCS was supplemented to the medium with the final concentration of 200 μM in YPD and 50 μM in V8.

**Cell growth assay.** Cryptococcal cells from overnight culture in YPD broth were centrifuged, washed twice with water, and resuspended in water. The cell concentration was normalized to the optical density of OD$_{600}$ = 1. Fifty microliters of each strain were added to 950 μL of the indicated liquid medium in a 24-well plate (#353047, Falcon). The growth of the cryptococcal strains (each with two replicates) was monitored at OD$_{600}$ hourly in the Cytation 5 multi-mode reader (BioTek Instrument) with double orbit shaking (at 365 cpm) at 30 °C. The reads for each well were fitted to a preloaded logistic model $\left( y = \frac{A1-A2}{1+\left(\frac{x}{x0}\right)^p} + A2 \right)$ in Origin software and the curated reads were simulated and graphed in the GraphPad Prism software.

**Mating, genetic crosses, and cell fusion assays.** Strains of α and **a** mating partners were crossed on V8 juice agar medium (pH 5 or 7) and incubated at 22 °C in dark for 2–3 weeks until adequate spores were produced. Spores were micro-manipulated using a dissecting microscope. The mating type of the germinated spores was determined by successful mating with either JEC20**a** or JEC21α reference strain. Genetic linkage between the presence of the drug marker and the observed mutant phenotype was established by analyzing the dissected spores as we described previously[57]. If the insertional mutant harbors only one copy of the T-DNA, which carries the hygromycin selection marker *HYG*, we would expect a 1:1 ratio of hygromycin-sensitive (HYG$^S$) and hygromycin-resistant (HYG$^R$) progeny. If that single insertion caused the non-filamentous phenotype, then all HYG$^R$ progeny would be non-filamentous while all HYG$^S$ progeny would be filamentous under a Znf2-inducing condition. In comparison,

when multiple T-DNAs inserted in the genome, the ratio between HYG$^S$ and HYG$^R$ meiotic progeny would be 1:3 (two unlinked insertions) or lower (>2 unlinked T-DNA insertions). For each genetic linkage assay, approximately 32 viable spores were analyzed, which gave a 97% confidence level of our analysis $\left( \text{confidence} = 100\% - \left(\frac{1}{2}\right)^{\log_2 \#\text{of spores}} \right)$.

To analyze the genetic relationship between Znf2 and Brf1, we crossed *brf1*::*NAT*/P$_{TEF1}$-*BRF1*-CBP-2xFLAG-*NEO* (α) with *znf2*::*NAT*/P$_{CTR4-2}$-3xFLAG-*ZNF2-NEO* (**a**) and examined the segregation of the genotypes (by diagnostic PCRs) and phenotypes in the progeny (Fig. 4a).

To determine the cell–cell fusion efficiency, mutants of the mating type α (*znf2*::*NAT*; *mat2*::*NAT*; *brf1*::*NAT*; and *snf5*::*NAT*) and the control strain (*prf1*::*NAT*) were collected and suspended in water to the same optical density (the calculated OD$_{600}$ = 0.3 × 10). Each mutant with NAT$^R$ was mixed with the mating type **a** tester strain with NEO$^R$ marker (strain YSB133) with equal volume[58]. Three microliters of cell mixture were spotted onto V8 medium and incubated at 22 °C in dark for 24 h before they were collected, plated (in serial dilution) onto YPD + NEO + NAT agar medium, and incubated at 30 °C for 3–5 days to select for fusion products.

**Databases and online tools.** We used FungiDB to acquire the gene/protein sequences or the normalized FPKM data of *C. neoformans* H99 genes, *S. cerevisiae* genes and genes in other fungi. Clustal Omega was used for protein multiple sequence alignment and phylogenetic tree analysis. The gene tree of *BRF1* (CNA02310) was retrieved from the Ensembl Fungi database. Simple Modular Architecture Research Tool (SMART) online server was used to analyze the domain layouts for proteins. IBS online illustrator tool was used to illustrate all the protein domain layouts[59].

**RNA extraction and RNA-seq.** RNAs were extracted from cells cultured on V8 medium at 22 °C in dark for 24 h and prepared for sequencing as previously described[19,20,60]. HiSeq Rapid 175bp pair-end RNA sequencing was performed at the Georgia genome sequencing facility. The low-quality bases of the raw reads were trimmed with a custom perl script as published before[60]. Tophat2 was used to map the processed reads to the reference genome. The program HTSeq-count and DESeq2 were used to count the reads and identify the differentially expressed genes. The raw sequencing reads were deposited at NCBI with the BioProject accession number PRJNA534125 and SRA file numbers from SRR8947060 to SRR8947075.

**Co-immunoprecipitation/mass spectrometry and co-IP/western.** Strains with tagged proteins were cultured in 15 mL YPD media overnight, washed twice with cold water, flash frozen in liquid nitrogen, and lyophilized. Lyophilized cells were broken into fine powder with silica beads in a Bullet Blender Blue® (Next Advance) without buffer for five cycles (60 s maximum blending followed by 90 s chilling on ice) and then with 1 mL of lysis buffer (50 mM Tris-HCl pH 8.0, 150 mM NaCl, 1 mM EDTA, 5 mM MgCl$_2$, 1 mM DTT, 10% glycerol, 0.5 mM PMSF, and cocktail protease inhibitors (#A32963, ThermoFisher)) for another five cycles. Cell lysates were centrifuged at 500 × *g* at 4 °C for 5 min. The supernatant was collected and centrifuged again at 10,000 × *g* at 4 °C for 20 min. The supernatant was then incubated with pre-washed anti-FLAG M2 agarose beads (#F2426, Sigma) at 4 °C on a rotator overnight. For immunoprecipitation with mCherry or mNeonGreen-tagged proteins, RFP-Trap®_MA or mNeonGreen-

Trap_MA beads from Chromotek (Germany) were used following the manufacturer's instructions.

After washing, immunoprecipitated protein samples were released from beads in the 2×SDS loading buffer (120mM Tris-HCl pH 6.8, 20% glycerol, 4% SDS, 0.04% bromophenol blue, and 10% β-mercaptoethanol) by boiling for 10 min.

For Co-IP/MS, protein samples were then loaded into pre-casted 4–12% SurePAGE™ Bis-Tris Gel (GeneScript), ran for 5 min, and visualized by Coomassie Blue staining. The total pulled down protein samples were excised from the gel and sent to the proteomics and mass spectrometry facility (https://pams.uga.edu/) at the University of Georgia for identification. When analyzing the results, we applied the following criteria to exclude nonspecific proteins: proteins from the negative control, proteins not shared between samples of the two $P_{TEF1}$-BRF1-CBP-2×FLAG strains, and non-nuclear proteins as Brf1 localizes to the nucleus.

For Co-IP/western, IP proteins samples from mNeonGreen-trap along with the whole protein extracts (WCE) were run in an SDS-PAGE gel (30 μL of WCE and IP), transferred onto PVSF membrane, and western-blotted with anti-FLAG M2 antibodies ((#F3165, Sigma).

**Chromatin immunoprecipitation, ChIP-qPCR, and ChIP-seq.** FLAG-tagged Znf2 strains (strain JL653 and JL665) (initial $OD_{600} = 0.2$) were incubated in 50mL YPD + BCS broth at 30 °C until the optical density reached $OD_{600} = 1$. Cells were then fixed in the medium with formaldehyde at 1% final concentration at 22 °C for 15 min with occasional swirling. Glycine with the final concentration of 0.125 M was added to quench the crosslinking at 22 °C for 5 min. The cells were then washed twice with cold sterile water and lyophilized. Lyophilized cells were broken as described above for Co-IP experiment with a different lysis buffer (50 mM HEPES-KOH pH 7.5, 140 mM NaCl, 1 mM EDTA, 1% Triton X-100, 0.1% NaDOC, 1 mM of PMSF, and proteinase inhibitor cocktail). The following steps were similar to what has been described previously[43,61–63] and briefly, the cell lysates were centrifuged at 8000 r.p.m. at 4 °C for 10 min to enrich nuclei. The released nuclei were resuspended in 350 μL of lysis buffer. The nuclear suspensions were sheared by sonication in a Diagenode Bioruptor™ for 25 cycles at 4 °C (30 s on, 30 s off; cycle numbers varied). The protein–chromatin complex was recovered from the supernatant after centrifugation at 14,000 r.p.m. at 4 °C for 10 min and the volume was brought to 1 mL. Fifty microliters of each protein–chromatin suspension was saved as Input DNA. The rest of the sample was added to 30 μL anti-FLAG M2 monoclonal antibodies on magnetic beads (#M8823, Sigma) to precipitate the FLAG-tagged protein. After incubation with antibodies at 4 °C overnight, the beads were washed twice with 1 mL ChIP lysis buffer, once in 1 mL ChIP lysis buffer containing 0.5 M NaCl, once in 1 mL ChIP wash buffer (10 mM Tris-HCl, pH 8.0; 0.25 M LiCl; 0.5% NP-40; 0.5% NaDOC; 1 mM EDTA), and once in 1 mL TE (10 mM Tris-HCl, pH 8.0; 1 mM EDTA), all at 4 °C. The immunoprecipitated Znf2-3xFLAG was eluted twice by adding 200 μL of TES buffer (10 mM Tris-HCl, pH 8.0; 10 mM EDTA; 1% SDS) and incubated at 75 °C for 15 min before the supernatant was transferred to a new tube. Twenty microliters of 5 M NaCl was added to each sample to de-crosslink at 65 °C for 4 h. Three hundred and fifty microliters of TES was added to each Input DNA sample before adding 20 μL of 5 M NaCl to reverse formaldehyde crosslinks. One microliter of RNAse A (10 μg/μL) was added and the sample was incubated at 37 °C for 30 min. Four microliters of Proteinase K (20 μg/μL) was then added to each sample and incubated at 45 °C for at least 1 h to digest all the proteins. Eventually, 2 μL of glycogen (molecular level, 20 mg/mL) was added to chromatin DNA samples. Ethanol of 2.5 volumes

was added to each sample to precipitate the glycogen tangled DNA. The precipitated DNA sample was then suspended in 80 μL of nuclease-free water.

For ChIP-qPCR, 3 μL of five times diluted DNA sample and 1 μL each from two primers (50 μM) were mixed with 5 μL of SYBR Green 2x qPCR premix reagents (Invitrogen). The qPCR reactions were carried out in a Realplex system (Eppendorf) with technical duplicates. The % Input is calculated as $2^{(-\Delta Ct\,[normalized\,ChIP])}$ where $\Delta Ct\,[normalized\,ChIP] = Ct\,[ChIP] - (Ct\,[Input] - log_2^{(input\,dilution\,factor)})$ as described previously[63].

The same ChIP samples for qPCRs were sent for DNA sequencing at Georgia Genomics Facility (UGA) on an Illumina NextSeq500 platform. Input from ChIP-seq experiments in WT strains was used as a background control for all samples. Differential peaks called using the findPeaks tool, part of them were annotated using ChipSeeker[64,65]. ChIP-seq have been submitted to the GEO database (accession # GSE137248).

**ATAC-seq and data analysis.** *C. neoformans* cells were cultured in 15 mL YPD liquid media at 30 °C overnight or on V8 medium (pH 7) at 22 °C in dark for 24 h. Cells were collected and washed two times with cold sterile water. Five hundred microliters of cold lysis buffer (15 mM Tris pH 7.5; 2 mM EDTA; 0.5 mM spermine; 80 mM KCl; 20 mM NaCl; 15 mM (or 0.1% v/v) β-me; 0.3% TrixtonX-100) and 200 μL (~1 PCR tube) of acid-washed glass beads (0.5mm, #9831 RPI) were added to the cell pellet of about $10^8$ cells in 1.7 mL Eppendorf tubes. Cells were broken in a cell disruptor (Scientific Industries, Inc., SI-D238) at 4 °C with maximum speed for 2 min and then spun down at $50 \times g$ at 4 °C for 2 min to remove the glass beads. The supernatant was transferred into a new Eppendorf tube and spun at $200 \times g$ for another 2 min to remove most of the broken cells and debris. The enrichment of nuclei in the supernatant was verified by microscopic observation and the nuclei were collected for making the ATAC-seq library.

ATAC-seq libraries were generated as follows. Briefly, about 0.2 million nuclei were incubated with Tn5 transposase preloaded with Illumina sequencing adapters at 37 °C for 30 min followed by purification of the DNA fragments by a reaction cleanup kit (Qiagen, # 28204). Libraries were PCR amplified for 10 cycles with Phusion polymerase (ThermoFisher Scientific, # F530L). Sequencing libraries were then cleaned with magnetic beads to remove free adapters and primer dimers (Beckman Coulter, #A63880). Libraries were mixed in equimolar ratios and pair-end sequenced by the Georgia Genomics Facility (UGA) on an Illumina NextSeq500 platform.

ATAC-seq reads were filtered for low-quality and short reads with Trim Galore (https://github.com/FelixKrueger/TrimGalore). Duplicate reads were removed using Picard MarkDuplicates tool (http://broadinstitute.github.io/picard/). Reads were aligned to the JEC21 reference genome (Refseq assembly GCF_000091045.1) with HISAT2 using non-spliced alignment and a maximum fragment length of 2000 bp. Peak calling was performed with MACS2 using a *q*-value of 0.01, extension size of 73, and shifting reads by 37 bp to center on the insertion site. MACS2 was also used to call differential peaks[66]. Differential peak files were first sorted by fold change and then combined into a single file. Genes nearby Znf2 peaks were annotated using ChipSeeker[64,65]. ATAC-seq enrichment was calculated across Znf2 peaks or differential peaks between WT and *brf1Δ* using the annotatePeaks function, part of the HOMER (Hypergeometric Optimization of Motif EnRichment) suite of informatics tools, with parameters -size 3000 and -ghist[67]. Heatmaps were generated in R using the pheatmap package version 1.0.12. Differential peaks from ATAC-seq were compared to each other or Znf2 peaks or to genes identified as being up-regulated in WT V8 from RNA-seq analysis using

BEDTools[68] intersect to isolate these genes from the JEC21 annotation file for determining genes associated with differential peaks. ATAC-seq have been submitted to the GEO database (accession # GSE137248).

**Statistics and reproducibility**. All the experiments were done at least in biological duplicates and showed consistent and reproducible results. For gene manipulations, multiple independently confirmed transformants were collected, and diagnostic PCR or genetic linkage analyses were used to confirm their genotypes. All confirmed mutants of the same genetic manipulation showed similar phenotypes. RNA-seq, ChIP-seq, and ATAC-seq were all done with two biological repeats and each sample had large number of cells (millions or more). More details can be referred to the corresponding sections.

**Reporting summary**. Further information on research design is available in the Nature Research Reporting Summary linked to this article.

### Data Accessibility

All data supporting the findings of the current study are available within the article and its Supplementary Information files or from the corresponding author upon request. All RNA-seq and DNA-seq data have been deposited in NCBI under the BioProject accession number PRJNA534125. ATAC-seq and ChIP-seq data are available at NCBI GEO database with access number GSE137248.

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

## Acknowledgements

We thank Dr. Karl Lechtreck from Department of Cellular Biology at UGA for the kind gift of the mNeonGreen plasmid, Dr. Chau-wen Chou at the UGA proteomics and mass spectrometry facility (https://pams.uga.edu/) for the mass spectrometry analysis of our Co-IP samples, Dr. Zefu Lu, Brigitte Hofmeister, and Dr. Bob Schmitz in Department of Genetics at UGA for the generous help on ATAC-seq experiments, Dr. Fanglin Zheng for help with yeast-two-hybrid experiments, and all Lin lab members for their helpful comments. This work was supported by National Institutes of Health (R01AI140719 to X.L. and R01GM132644 to Z.A.L.), the American Cancer Society (RSG-14-184-01-DMC to Z.A.L.), NSF GRFP (NSF# 1443117 to A.R.F.), and the University of Georgia (startup fund to X.L.). Dr. Xiaorong Lin holds an Investigator Award in the Pathogenesis of Infectious Disease from the Burroughs Wellcome Fund (1012445). The funders had no role in study design, data collection and interpretation, or the decision to submit the work for publication.

## Author contributions

J.L., Z.A.L., and X.L. designed the study. J.L., A.R.F., and Y.Z. conducted the experiments. Y.Z. and E.Y. analyzed the DNA sequencing data; Y.Z. and J.L. analyzed RNA sequencing data; A.R.F. and Z.A.L. analyzed the ATAC and ChIP-DNA sequencing data. J.L. and X.L. wrote the manuscript with contributions from A.R.F. and Z.A.L.

## Competing interests

The authors declare no competing interests.
