## [Peer Review File · Communications Biology]

Reviewers' comments:

Reviewer #1 (Remarks to the Author):

This article focuses on the function of the SWI/SNF chromatin remodelling complex in the Basidiomycetes yeast *Cryptococcus neoformans*. The rationale is clear: The yeast to hyphae switch in *C. neoformans* is regulated by the zinc-finger transcription factor Znf2. Deletion of Znf2 blocks the yeast to hyphae transition while overexpression of Znf2 triggers the yeast to hyphae switch regardless of the growth conditions. The authors performed a genetic screen to identify proteins working in concert or downstream of Znf2. This genetic screen led to the following discoveries:

1. In *C. neoformans* the SWI/SNF component SNF5 is required for the yeast to hyphae switch. This is an important piece of information but not surprising per se given that (as stated by the authors) Snf5 is critical for cellular differentiation in all organisms tested, including ascomycetous and basidiomycetous fungi
2. BRF1, a basidiomycete-specific protein, is a component of the SWI/SNF complex and consequently regulates hyphal differentiation. This is where the novelty of the paper lies as until now this was an uncharacterized novel protein. The novelty of these findings is reduced by the observation that BRF-1 contains an ARID domain. From yeast to human, SWI/SNF complexes are known to contain ARID-containing proteins
3. By combining RNA-seq, ATAC-seq and ChIP analyses, the authors demonstrate that Brf1 is required for chromatin remodelling of Znf2 promoter and promoters of its target genes and it is required for full association of the transcription factor Znf2 with its downstream targets.

This is a complete project: starting from the identification of novel factors to the characterisation of the identified genes. State-of-the-art technologies are used. In general the article is well written and logical. Genome-wide datasets are performed in biological duplicates.

I have a few suggestions:

1. The RNA-seq and ATAC-seq datasets should be presented more comprehensively.
2. RNA-seq of wt, *brf1Δ*, *brf1Δ/BRF1oe*, and *ZNF2oe* strains cultured under filamentation-repressing YPD medium and filamentation-inducing conditions has been performed. It is important to know the genes differentially expressed in *brf1Δ* cells compared to WT in filamentation-repressing media? Does Brf1 have a Znf2-independent role in these conditions?
3. Likewise the ATAC-seq dataset should be presented more comprehensively: What are the differences in chromatin accessibility in WT cells grown in repressing and inducing-filamentation conditions? The authors stated ". We identified 39 regions that displayed reduced accessibility in the *brf1Δ* strain". What are those regions? Is the ATAC-seq profiling in *brf1Δ* strains similar in repressing and inducing-filamentation conditions?
4. The IP/MassSpec hits should be confirmed by IP/Western
5. Throughout the manuscript, the authors should make clear that ARID-containing proteins are known components of the SWI/SNF remodelling complex in many organisms. This has been mentioned in the discussion only briefly.
6. I would shorten the results section discussing the nature of SWI/SNF and RSC complex in *C. neoformans* versus *S. cerevisiae* and this is lengthy and not central to the paper.

Minor comments:

1. Line 231: CoIP, need to clarify that Brf1-proteins were identified by Mass Spectrometry.

Reviewer #2 (Remarks to the Author):

This paper has an in-depth study of SWI/SNF complexes that act in concert with Znf2, which plays a major role in the filamentous growth in *C. neoformans*. In contrast to Ascomycota, authors provide evidence that RSC or SWI/SNF complexes may not be essential for the growth of *C. neoformans*. Both

SWI/SNF complex and Znf2 are important for yeast-hypha differentiation, and both are required for transcriptional induction of CFL1 which is one of the major downstream factors of Znf2.

In particular, it is interesting that screening of a large number of mutants involved in filamentous growth in the ZNF2 overexpression and linking the key subunit Snf5 of the SWI/SNF chromatin remodeling complex with Brf1, which has an ARID domain, in a remarkable relationship with previous findings of yeast-to-hyphal differentiation in terms of genetics.

However, as commented in discussion, it is somewhat unfortunate that the authors could not provide evidence that Znf2 and SWI/SNF interact physically. If the weak or transient interaction were not proven by Co-IP as described in the text, it would be considered that the possibility to prove it by using more rough way, such as yeast-two-hybrid system.

Line 116: Please describe the meaning of the criterion excluding the mutants showing non-nuclear mCherry-Znf2. Subsequent studies of regulators that inhibit nuclear translocation of the transcription factor Znf2 seem to be interesting to understand the overall mechanisms of regulation of Znf2-dependent filamentous growth.

Line 298-311: The fold-change scores do not seem to be correlated with Figure 4C because induction or reduction does not appear on the graph. It would be better to be shown by using volcano plot or using different color codes for the fold-changes. The authors also should provide or explain the P values or the cutoff range of the P value for the DEG analysis.

Line 335-354: The ATAC-seq was performed to demonstrate the authors' hypothesis for the Brf1-dependent chromatin modification of the Znf2 target as well as the promoter region of ZNF2. The 10 Brf1-dependent peaks including CFL1 locus should contain the data of the promoter region of ZNF2. The meaning of the control (CNH02240) should also be described in the text or figure legend.

Line 371: Figure 5E and S6 require statistics on the %input values and statements about the meaning of error bars implicating biological or technical replicates.

Line 105: Genotype display needs to be unified. Depending on the plasmid name, the integrated gene should be dashed to the promoter and terminator (promoter-coding region-terminator). Some genotype names do not have dashes between promoter, coding sequence, and terminator.

Line 502: Please explain the CBP protein. (CREB binding protein?)

Table 1,2 and Table S3: Please put the explanation about the color codes below the tables.

Reviewer #3 (Remarks to the Author):

The authors perform a forward genetic screen to identify genes that are required for filamentation of Cn under conditions of Znf2 expression, which is the downstream transcription factor and primary driver of filamentation. The screen identified several loci that impact the function of Znf2 by different mechanisms. Of particular interest were two genes, SNF5 and BRF1, in which Znf2 was localized to then nucleus, but was unable to promote filamentation.

Analysis of Snf5 and Brf1 sequence revealed homology to SWI/SNF components and a ARID-domain containing protein, respectively. The latter appears to be specific to the basidiomycetes. Deletion of

either gene impairs filamentation and cell-cell fusion, and the double deletion mutant is not additive, suggesting the Snf4 and Brf1 function together to regulate Znf2-dependent gene expression.

The known role of Snf5 in chromatin modification led the authors to hypothesize that the function of Snf5, in conjunction with Brf1, is to regulate access of Znf2 to promoter regions through nucleosome repositioning. The authors go on to demonstrate that Brf1 is in a complex with Snf5 by IP, and does not interact with the related RSC complex, suggesting the Brf1 is indeed a basidiomycete specific component of the SWI/SNF complex. The study also revealed sharing of subunits between SWI/SNF and RSC in *Cn* that do not occur in *S. cerevisiae*.

Brf1 is necessary for transcriptional induction of Znf2 in response to filamentation cues, and for the transcriptional induction of filamentation promoting genes. In the absence of Brf1, Znf2 interaction with target promoters is reduced, and ATAC-seq demonstrated that a subset of closed promoters in the Brf1 deletion mutant overlapped with filamentation-promoting genes, one of which is Cfl1, known to be important in filamentation.

Overall the manuscript presents complete story arc from forward genetic identification of Znf2-collaborators/regulators through a mechanistic investigation of their function. The manuscript provides a first description of a novel, basidiomycete specific component of the SWI/SNF complex and demonstrates its specificity and distinction from the related RSC complex. In addition, the ATAC-seq data provide additional mechanistic insight into the regulation of Znf2-target genes at the level of promoter accessibility. Overall, this is a complete and rigorous analysis that is of interest to fungal biology and developmental biology broadly.

Critiques:

1. A subset of Znf2 targets are regulated by Brf1 - is there any common feature of these promoters? Is the znf2 consensus known, and are there multiple binding sites?
2. The authors report a defect in cell fusion in the Brf1 deletion mutant. This is likely upstream of Znf2 - is there any clue from the ATAC-seq data to suggest Znf2-independent targets that regulate cell fusion?
3. There are some EM and structural models of the *Sc* SWI/SNF complex. based on homologous proteins, do you have an idea which subunits would potentially interact with Bfr1? Where do you see it fitting into the complex? Do you have data that Snf5 and Brf1 interact directly with one another, or do you think that Brf1 is associating with the complex through other subunits? Have you performed the reverse IP, and does Brf1 come down with tagged Snf5?

Reviewers' comments:

Reviewer #1 (Remarks to the Author):

This article focuses on the function of the SWI/SNF chromatin remodelling complex in the Basidiomycetes yeast *Cryptococcus neoformans*. The rationale is clear: The yeast to hyphae switch in *C. neoformans* is regulated by the zinc-finger transcription factor Znf2. Deletion of Znf2 blocks the yeast to hyphae transition while overexpression of Znf2 triggers the yeast to hyphae switch regardless of the growth conditions. The authors performed a genetic screen to identify proteins working in concert or downstream of Znf2. This genetic screen led to the following discoveries:

1. In *C. neoformans* the SWI/SNF component SNF5 is required for the yeast to hyphae switch. This is an important piece of information but not surprising per se given that (as stated by the authors) Snf5 is critical for cellular differentiation in all organisms tested, including ascomycetous and basidiomycetous fungi
2. BRF1, a basidiomycete-specific protein, is a component of the SWI/SNF complex and consequently regulates hyphal differentiation. This is where the novelty of the paper lies as until now this was an uncharacterized novel protein. The novelty of these findings is reduced by the observation that BRF-1 contains an ARID domain. From yeast to human, SWI/SNF complexes are known to contain ARID-containing proteins
3. By combining RNA-seq, ATAC-seq and ChIP analyses, the authors demonstrate that Brf1 is required for chromatin remodelling of Znf2 promoter and promoters of its target genes and it is required for full association of the transcription factor Znf2 with its downstream targets.

This is a complete project: starting from the identification of novel factors to the characterisation of the identified genes. State-of-the-art technologies are used. In general the article is well written and logical. Genome-wide datasets are performed in biological replicates.

We thank the reviewer for the kind summary of this study.

I have a few suggestions:

1. The RNA-seq and ATAC-seq datasets should be presented more comprehensively.

We repeated ATAC-seq to obtain more reads and further analyzed our RNA-seq and ATAC-seq datasets. In this revised manuscript, we have supplemented the lists of differentially expressed genes (DEGs, $|\log_2(\text{fold change})| > 2$ and $\text{FDR} < 0.05$) as Supplementary Table 4, in which we included DEGs for *brf1*Δ and *znf2*Δ strains cultured under filamentation-suppressing and filamentation-inducing conditions. The original Figure 4C only listed the selected genes in the pheromone pathway. During the revision process, we replaced the old Fig. 4c with a new one where we plotted all the differentially regulated genes in *brf1*Δ and *znf2*Δ strains cultured under the filamentation-inducing condition (V8) using a volcano plot. We also made a similar volcano plot of all the differentially expressed genes in the *brf1*Δ and *znf2*Δ strains cultured under the filamentation-suppressing condition (YPD) and added the information as Supplementary Fig. 6. In both Fig. 4c and Supplementary Fig. 6, the significantly differentially expressed genes shared between the *brf1*Δ and the

znf2Δ strains are colored orange. As shown in the new Fig. 4c, more than 95% of the overlapping genes (in orange) were down regulated upon deletion of *BRF1* or *ZNF2*, suggesting that Brf1 and Znf2 activate these genes during hyphal differentiation. A few shared DEG genes in the pheromone pathway or those known to be associated with filamentation were indicated in the new Fig. 4c. Likewise, we performed additional ATAC-seq replicate experiments and re-analyzed differential enrichment data. We have expanded our description of these data in the results section. We now provide the lists of differential peaks between WT and various mutants and between filamentation-inducing and -repressing conditions for wild type as Supplementary Table 8. In addition, to better probe the relationship between Brf1 and Znf2, we carried out new ChIP-DNA seq experiments to determine binding sites for Znf2, and we compared global Znf2 binding data with ATAC-seq data. We added new panels Fig. 5b and 5e and we updated panels 5c, 5d, and 5f to reflect these changes. (lines 347-433).

2. RNA-seq of wt, *brf1Δ*, *brf1Δ/BRF1oe*, and *ZNF2oe* strains cultured under filamentation-repressing YPD medium and filamentation-inducing conditions has been performed. It is important to know the genes differentially expressed in *brf1Δ* cells compared to WT in filamentation-repressing media? Does Brf1 have a Znf2-independent role in these conditions?

We agree with the reviewer that examining the differentially expressed genes between *brf1Δ* and wild type under filamentation-suppressing conditions will be informative. In the revised manuscript, we have added the new Supplementary Table 4, which includes the DEGs for both the *brf1Δ* mutant and the *znf2Δ* mutant comparing to the wild type in filamentation-suppressing and filamentation-inducing conditions. Under the filamentation-suppressing condition used (YPD media), deletion of *ZNF2* altered the transcript level of 378 genes and deletion of *BRF1* affected the transcript level of 85 genes using the criterion: $|\log_2FC| \geq 2$ and $FDR < 0.05$. Only 19 of the differentially expressed genes were shared between these two mutants under the filamentation-suppressing condition (Supplementary Fig. 6). The shared 19 DEGs did not yield any significant enrichment of GO terms in biological processes. The DEGs specific to the *brf1Δ* mutant under the filamentation-suppression condition are enriched in the categories of membrane transporters for anion, ion and amino acids, and in oxidation-reduction process. The decreased expression of ion transporters in the *brf1Δ* mutant might be responsible for the increased sensitivity of this mutant towards osmotic stress caused by 1M NaCl or KCl salt (Fig. 2b) or the poor utilization of raffinose (Fig. 2c). By contrast, the *znf2Δ* mutant did not show any altered sensitivity to NaCl/KCl or raffinose comparing to the wild type control. The data suggest that Brf1 has a role in certain stress responses that is independent of Znf2 (Fig. 2b and 2c). We briefly described the information in the revised manuscript (Line 202-211).

3. Like-wise the ATAC-seq dataset should be presented more comprehensively: What are the differences in chromatin accessibility in WT cells grown in repressing and inducing-

filamentation conditions? The authors stated “. We identified 39 regions that displayed reduced accessibility in the *brf1Δ* strain”. What are those regions? Is the ATAC-seq profiling in *brf1Δ* strains similar in repressing and inducing-filamentation conditions?

As mentioned in our response to question #1, we added additional ATAC seq replicates and carried out additional analysis of differential accessibility for all strains and conditions tested. Peak information and differential peaks are now reported in Supplementary Table 6 and Supplementary Table 8. In particular, we annotated genes located near regions that displayed altered accessibility in the *brf1Δ* mutant cultured under filamentation-inducing conditions (Supplementary Table 7 and Supplementary Fig. 8). Both *CFL1* and *ZNF2* require Brf1 for accessibility (as shown in Fig. 5f). We also included information about differential accessibility for the wild type grown in filamentation-inducing and -repressing conditions. Only 421 regions show differential accessibility in wild type grown in YPD and V8, whereas deletion of *BRF1* results in altered accessibility at 1312 sites. 90% of differential peaks in YPD versus V8 overlapped the differential peaks identified by comparing peaks in WT V8 versus the *brf1Δ* strain; however, as expected, Brf1 has a larger impact on chromatin structure. As expected, the *CFL1* locus (the filamentation marker) only showed up in the filamentation-inducing condition even in the wild type as this region is closed when cells are cultured under the filamentation-suppressing condition.

4. The IP/MassSpec hits should be confirmed by Ip/Western

The co-immunoprecipitation experiments using the tagged Brf1 protein as the bait were carried out in duplication. Only the shared hits between the duplicates and those that were absent from the non-tagged control were considered as potential positive hits. Several proteins that were identified as potential Brf1 partner proteins were in the SWI/SNF family (some are unique to SWI/SNF complex and some are shared also by the RSC complex). Similarly, we performed co-immunoprecipitation experiments using the RSC-complex specific subunit Sfh1 protein as the bait. Many components shared by the SWI/SNF complex and the RSC complex, and some components unique to the RSC complex were identified (Table 1 and Supplementary Table 3). Thus, we are confident about the listed hits identified from the co-immunoprecipitation experiments. To further confirm the interactions between Brf1 and Snf5, we conducted a reciprocal Co-IP/MS experiment with the Snf5-mNeonGreen protein as the bait in a Brf1-FLAG strain. We first tagged Snf5 with mNeonGreen at its native locus. However, we could not detect Snf5-mNeonGreen based on fluorescence or western blot (data not shown) possibly because of its low expression level when *SNF5* was driven by its own promoter (Fig. 3d). We then constructed the Snf5-mNeonGreen protein driven by *GPD1* promoter Snf5-mNeonGreen is functional given that it can complement the *snf5Δ* mutant phenotype. We then did Co-IP/Western as suggested by the reviewer. As shown in the newly added Supplementary Fig. 4c, Snf5-mNeonGreen brought down Brf1-FLAG. Thus, the original and the newly acquired evidence strongly supports the interaction between Brf1 and Snf5.

5. Throughout the manuscript, the authors should make clear that ARID-containing proteins are known components of the SWI/SNF remodelling complex in many organisms. This has been mentioned in the discussion only briefly.

We thank the reviewer for this suggestion. The brief discussion of ARID domain-containing proteins in the original manuscript reflects how this study was developed over the course of several years. We first did the forward genetic screen, identified the Snf5 and Brf1, speculated that they function in the same complex given the similar phenotypes of the *brf1*Δ and *snf5*Δ mutants, and then realized later that Brf1 might serve as the ARID domain protein in the SWI/SNF complex. That revelation was not as apparent to us as it might be described in the manuscript as there were only few studies about ARID domain proteins in the SWI/SNF complex in the large body of literature on the SWI/SNF complex. Because there are multiple ARID domain proteins in the genome, we then confirmed that Brf1, comparing to other ARID domain proteins, has a specific role to filamentation by examined the disruption of these genes on cryptococcal morphogenesis (line 231-237). In addition to the discussion session, we also added a sentence about ARID domain proteins in the complex in the result session where we confirmed that Brf1 works in the complex (267-269).

6. I would shorten the results section discussing the nature of SWI/SNF and RSC complex in *C. neoformans* versus *S. cerevisiae* and this is lengthy and not central to the paper.

We thank the reviewer for this suggestion. We shortened this comparative analysis of SWI/SNF and RSC complex in *C. neoformans* and *S. cerevisiae* in the result session. Although we agree with the reviewer that this is not central point of this manuscript, the differences of this highly conserved complex that we identified in the two divergent fungal species (ascomycete and basidiomycete) might be valuable to other researchers. So we decided to move some of the analyses to the discussion session in the revised manuscript for readers who may find the information useful.

Minor comments:

1. Line 231: CoIP, need to clarify that Brf1-proteins were identified by Mass Spectrometry.

Thank you. We have added the information in the main text.

Reviewer #2 (Remarks to the Author):

This paper has in-depth study of SWI/SNF complexes that act in concert with Znf2, which plays a major role in the filamentous growth in *C. neoformans*. In contrast to Ascomycota, authors provide evidence that RSC or SWI/SNF complexes may not be essential for the growth of *C. neoformans*. Both SWI/SNF complex and Znf2 are important for yeast-hypha differentiation, and both are required for transcriptional

induction of CFL1 which is one of the major downstream factors of Znf2.

In particular, it is interesting that screening of a large number of mutants involved in filamentous growth in the ZNF2 overexpression and linking the key subunit Snf5 of the SWI/SNF chromatin remodeling complex with Brf1, which has an ARID domain, in a remarkable relationship with previous findings of yeast-to-hyphal differentiation in terms of genetics.

We thank the reviewer for the kind summary of the work.

However, as commented in discussion, it is somewhat unfortunate that the authors could not provide evidence that Znf2 and SWI/SNF interact physically. If the weak or transient interaction were not proven by Co-IP as described in the text, it would be considered that the possibility to prove it by using more rough way, such as yeast-two-hybrid system.

We thank the reviewer for this suggestion. We tested the yeast-two-hybrid system. However, when we fused the transcription factor Znf2 with the DNA-binding domain (DB domain) and expressed it in the Y2H *Saccharomyces* strain as the bait, it showed auto-activation. We think that autoactivation is likely due to Znf2 protein itself carrying its own activation domain. Znf2's own activation domain recruits the RNA polymerase to the reporter gene because the fused DB domain binds to the *GALI* promoter of the reporter gene, which consequently causes autoactivation. The C-terminus of the Znf2 protein indeed has two strong acidic regions, which often serve as activation domains. To remove the potential activation domain within Znf2, we made two truncated versions of Znf2: one with the most-terminal acidic domain truncated, and the other with both acidic domains truncated. Unfortunately, we still detected autoactivation with both versions of the truncated Znf2. We then tried fuse Znf2 with the AD domain. Unfortunately, co-expressing Znf2 fused with the AD domain and the vector with the empty BD in yeast also autoactivates the reporter gene. Therefore, we could not test the interaction between Znf2 and Brf1 via this approach. So far, we could not prove direct interaction between Znf2 and the SWI/SNF complex based on our Co-IP and Y2H experiments. That said, we cannot rule out the possibility that Znf2 could directly interact with the SWI/SNF complex through other subunits or through histone modifications. We added a brief discussion of the result in the revised manuscript. (lines 456-467).

Line 116: Please describe the meaning of the criterion excluding the mutants showing non-nuclear mCherry-Znf2. Subsequent studies of regulators that inhibit nuclear translocation of the transcription factor Znf2 seem to be interesting to understand the overall mechanisms of regulation of Znf2-dependent filamentous growth.

We agree with the reviewer that the mutants with mis-localized Znf2 signal are interesting as they could help us understand how Znf2 protein is regulated through trafficking, post-translational modification, and/or protein stability. Two of those mutants showed T-DNA insertions in the P_{CTR4-2}-mCherry-ZNF2 construct, which disrupted the proper nuclear localization of Znf2 (Supplementary Table S1). However, we do not know where T-DNA insertions are in the remaining mutants

with mislocalized Znf2-mCherry signal. We plan to pursue these mutants in a separate study (the junior graduate student Amber Matha just took up the torch to continue this project).

For this current study, we focused on mutants that have no defect in Znf2's subcellular localization. The Znf2 protein is expressed and it is localized properly in the nucleus. This allows us to focus on factors that are required for Znf2 protein to exert its function in the nucleus. We clarified our logic in the revised manuscript as follows: "Fifty-eight of these mutants showed nuclear mCherry-Znf2 localization while 26 had altered subcellular distribution of the fluorescence signal. The mutants with mis-localized Znf2 signal will help understand how Znf2 protein is regulated through trafficking, post-translational modification, and/or stability. In this study, we set to identify factors that are required for nuclear localized Znf2 protein to exert its function. Thus, we decided to follow up the 58 mutants with Znf2 expressed and properly localized to the nucleus..." (Lines 120-126)

Line 298-311: The fold-change scores do not seem to be correlated with Figure 4C because induction or reduction does not appear on the graph. It would be better to be shown by using volcano plot or using different color codes for the fold-changes. The authors also should provide or explain the P values or the cutoff range of the P value for the DEG analysis.

We thank the reviewer for the suggestions. We have supplemented the lists of DEG analysis for the *brf1*Δ and the *znf2*Δ mutants under the filamentation-inducing condition and the filamentation-suppressing condition in the Supplementary Table 4. We have also remade the Fig. 4c using volcano plots and showed all the differentially regulated genes in *brf1*Δ and *znf2*Δ strains in the filamentation-inducing condition (V8). The shared significantly differentially expressed genes ($|\log_2(\text{fold change})| > 2$ and $\text{FDR} < 0.05$) between the *brf1*Δ and the *znf2*Δ mutant strains are colored in orange in the graphs. We made similar volcano plots for all the differentially regulated genes in *brf1*Δ and *znf2*Δ strains in the filamentation-suppressing condition (YPD). This is added as Supplementary Fig. 6. Please also see our response to the question #1 of reviewer #1 for more details.

Line 335-354: The ATAC-seq was performed to demonstrate the authors' hypothesis for the Brf1-dependent chromatin modification of the Znf2 target as well as the promoter region of ZNF2. The 10 Brf1-dependent peaks including CFL1 locus should contain the data of the promoter region of ZNF2. The meaning of the control (CNH02240) should also be described in the text or figure legend.

As mentioned in our response to reviewer 1, we added additional ATAC-seq replicates and ChIP-seq data for Znf2 to better probe the relationship between Brf1 and Znf2. We have updated the information for the differential peaks in various mutants including the *brf1*Δ strain under the filamentation-inducing condition in the Supplementary Table 7. ZNF2 is one of the 41 Brf1-dependent peaks and we have added the information in the updated Fig. 5e and in the main text (line 429-

431). The control gene CNA07690 encodes a hypothetical protein and its promoter region does not require Brf1 for open chromatin based on our ATAC-seq data. We have also updated the information about the control gene in the figure legends as suggested by the reviewer.

Line 371: Figure 5E and S6 require statistics on the %input values and statements about the meaning of error bars implicating biological or technical replicates.

The standard derivation error bars represent the data from two technical replicates of each biological duplicate. We have updated the corresponding figure legends and the methods in the text.

Line 105: Genotype display needs to be unified. Depending on the plasmid name, the integrated gene should be dashed to the promoter and terminator (promoter-coding region-terminator). Some genotype names do not have dashes between promoter, coding sequence, and terminator.

We appreciate the carefulness of the reviewer. We have made the genotype display uniformly (strain genotypes in Supplementary Table 11 and the plasmids listed in the Supplementary Table 12).

Line 502: Please explain the CBP protein. (CREB binding protein?)

CBP stands for calmodulin-binding peptide. CBP is used as one of the tandem tags here as described previously by other researchers in the *Cryptococcus* field^{1,2,3}. Our proteins are tagged with CBP-2xFLAG, but we only used the FLAG epitope for the Co-IP experiments.

Table 1,2 and Table S3: Please put the explanation about the color codes below the tables.

We thank the reviewer for the suggestion. We consistently use the same set of color codes for the SWI/SNF and RSC subunits in all our figures and tables, including Fig. 3d, Fig. 3g, Table 1, Table 2, and Supplementary Table 3. The SWI/SNF-specific subunits are colored in blue, the RSC-specific subunits are colored in yellow, and the SWI/SNF and RSC shared subunits are colored in green. We have explained the color codes both in the table notes and in the corresponding figure legends.

Reviewer #3 (Remarks to the Author):

The authors perform a forward genetic screen to identify genes that are required for filamentation of *Cn* under conditions of Znf2 expression, which is the downstream transcription factor and primary driver of filamentation. The screen identified several loci that impact the function of Znf2 by different mechanisms. Of particular interest were two genes, SNF5 and BRf1, in which Znf2 was localized to then nucleus, but was unable to promote filamentation.

Analysis of Snf5 and Brf1 sequence revealed homology to SWI/SNF components and a

ARID-domain containing protein, respectively. The latter appears to be specific to the basidiomycetes. Deletion of either gene impairs filamentation and cell-cell fusion, and the double deletion mutant is not additive, suggesting the Snf4 and Brf1 function together to regulate Znf2-dependent gene expression.

The known role of Snf5 in chromatin modification led the authors to hypothesize that the function of Snf5, in conjunction with Brf1, is to regulate access of Znf2 to promoter regions through nucleosome repositioning. The authors go on to demonstrate that Brf1 is in a complex with Snf5 by IP, and does not interact with the related RSC complex, suggesting the Brf1 is indeed a basidiomycete specific component of the SWI/SNF complex. The study also revealed sharing of subunits between SWI/SNF and RSC in *Cn* that do not occur in *S. cerevisiae*.

Brf1 is necessary for transcriptional induction of Znf2 in response to filamentation cues, and for the transcriptional induction of filamentation promoting genes. In the absence of Brf1, Znf2 interaction with target promoters is reduced, and ATAC-seq demonstrated that a subset of closed promoters in the Brf1 deletion mutant overlapped with filamentation-promoting genes, one of which is *Cfl1*, known to be important in filamentation.

Overall the manuscript presents complete story arc from forward genetic identification of Znf2-collaborators/regulators through a mechanistic investigation of their function. The manuscript provides a first description of a novel, basidiomycete specific component of the SWI/SNF complex and demonstrates its specificity and distinction from the related RSC complex. In addition, the ATAC-seq data provide additional mechanistic insight into the regulation of Znf2-target genes at the level of promoter accessibility. Overall, this is a complete and rigorous analysis that is of interest to fungal biology and developmental biology broadly.

We thank the reviewer for the appreciation of this study.

Critiques:

1. A subset of Znf2 targets are regulated by Brf1 - is there any common feature of these promoters? Is the znf2 consensus known, and are there multiple binding sites?

The consensus binding sites for this transcription factor is unknown. During the revision process, we sent our Znf2 ChIP DNA used in our ChIP-qPCR for DNA sequencing. We tried to predict the consensus binding motifs of Znf2 from our ChIP-seq data. However, the motif predictions varied among samples. One possible reason is that the ChIP-qPCR samples that we sent out for sequencing had long DNA fragments (~300bp median), which is not ideal for motif prediction. Our experience of predicting consensus of transcription factors using promoter regions of DEGs was not consistent or reliable. We thus decided not to mention this in this current study to avoid disseminating potentially erroneous information.

2. The authors report a defect in cell fusion in the Brf1 deletion mutant. This is likely upstream of Znf2 - is there any clue from the ATAC-seq data to suggest Znf2-independent targets that regulate cell fusion?

We agree with the reviewer that Brf1 regulates multiple processes during sexual reproduction, including cell fusion and initiation of hyphal growth. Our RNA-seq data indicate that Brf1 not only is required for transcription induction of the *ZNF2* gene, but it also controls the transcript levels of many genes in the cell fusion process such as the pheromone exporter gene *STE6* and the pheromone downstream transcription factor *MAT2* that work genetically upstream of *ZNF2* (Fig. 4c and Supplementary Table 4). Our ATAC-seq data also indicate that Brf1 regulates several processes of the sexual development. In Supplementary Table 7, we showed that 41 regions displayed severely reduced chromatin accessibility upon *BRF1* deletion. These include the promoter regions of *ZNF2* and some Znf2 targets like *CFL1*. The promoters of *SXII α* and *CRG1* also showed reduced accessibility. *CRG1* is known to regulate the pheromone sensing and cell fusion processes. *SXII α* is known to regulate hyphal growth after cell fusion during bisexual mating. Therefore, both transcriptome and ATAC-seq results suggest that Brf1 functions at multiple stages of sexual development in *Cryptococcus neoformans*.

3. There are some EM and structural models of the Sc SWI/SNF complex. based on homologous proteins, do you have an idea which subunits would potentially interact with Bfr1? Where do you see it fitting into the complex? Do you have data that Snf5 and Brf1 interact directly with one another, or do you think that Brf1 is associating with the complex through other subunits? Have you performed the reverse IP, and does Brf1 come down with tagged Snf5?

The SWI/SNF complex in Sc and Cn should be quite different, as we discussed in the manuscript. (1) Snf5 in Cn is 190% larger in mass than the Snf5 subunit in Sc. (2) Brf1 in *Cryptococcus* is completely different from Swi1 in *Saccharomyces*. The only conserved feature is the ARID domain (127 aa out of the 1033 aa Brf1 or 87 aa/1314 aa of Swi1). Even for the 12% portion of the proteins, the amino acid sequence homology is only 24% identical between Swi1 and Brf1 (protein percent identity calculated by Clustal2.1). (3) In Sc, the *swi1* and *snf5* mutants exhibit overlapping but distinct phenotypes, suggesting that Swi1 and Snf5 are likely in different modules. By contrast, *brf1 Δ* and *snf5 Δ* mutants are almost phenotypically identical in Cn, suggesting that they could be in the same module during assembly. Due to these differences, it would be challenging to use modeling to assemble the complex. The differences in the SWI/SNF complex between Sc and Cn are discussed in the manuscript (line 229-232, line 278-289, and line 484-498).

In this revision, we performed reciprocal Co-IP/western to further confirm the interaction between Brf1 and Snf5. We first tagged Snf5 with mNeonGreen at its native locus. However, we could not detect Snf5-mNeonGreen based on fluorescence or western blot (data not shown) possibly because of the low expression level when *SNF5* is driven by its own promoter (Fig. 3d). Then we constructed the Snf5-

mNeonGreen protein driven by the constitutively active *GPD1* promoter. The mNeonGreen tagged Snf5 is functional given that it complement the *snf5* Δ mutant phenotype. We transformed *SNF5*-mNeonGreen into a strain with FLAG-tagged Brf1. Then we used mNeonGreen trap for immunoprecipitation and used anti-FLAG for western. As shown in the newly added Supplementary Fig. 4c, Snf5 indeed pulled down Brf1. Thus, the original and the newly acquired data strongly support the interaction between Brf1 and Snf5.

References cited in this letter

1. Dumesic PA, *et al.* Product binding enforces the genomic specificity of a yeast polycomb repressive complex. *Cell* **160**, 204-218 (2015).
2. Chun CD, Brown JCS, Madhani HD. A major role for capsule-independent phagocytosis-inhibitory mechanisms in mammalian infection by *Cryptococcus neoformans*. *Cell host & microbe* **9**, 243-251 (2011).
3. Homer CM, *et al.* Intracellular Action of a Secreted Peptide Required for Fungal Virulence. *Cell host & microbe* **19**, 849-864 (2016).

REVIEWERS' COMMENTS:

Reviewer #1 (Remarks to the Author):

The authors addressed all issues in revision really nicely.

Reviewer #2 (Remarks to the Author):

I agree with the revisions made to the manuscript.

Reviewer #3 (Remarks to the Author):

All concerns were addressed adequately. This is an excellent body of work demonstrating important regulation of mating and sexual development by a *C. neoformans* SWI/SNF complex.